# Population dynamics of foxes during restricted-area culling in Britain: Advancing understanding through state-space modelling of culling records

Tom A. Porteus[1¤]*, Jonathan C. Reynolds[2], Murdoch K. McAllister[3]

**1** Department of Zoology, University of British Columbia, Vancouver, BC, Canada, **2** Game & Wildlife Conservation Trust, Fordingbridge, Hampshire, United Kingdom, **3** Institute for the Oceans and Fisheries, University of British Columbia, Vancouver, BC, Canada

¤ Current address: Game & Wildlife Conservation Trust, Fordingbridge, Hampshire, United Kingdom
* tporteus@gwct.org.uk

**Data Availability Statement:** All relevant data are within the manuscript and its Supporting Information files.

## Abstract

Lethal control is widely employed to suppress the numbers of target wildlife species within restricted management areas. The success of such measures is expected to vary with local circumstances affecting rates of removal and replacement. There is a need both to evaluate success in individual cases and to understand variability and its causes. In Britain, red fox (*Vulpes vulpes*) populations are culled within the confines of shooting estates to benefit game and wildlife prey species. We developed a Bayesian state-space model for within-year fox population dynamics within such restricted areas and fitted it to data on culling effort and success obtained from gamekeepers on 22 shooting estates of 2 to 36 km². We used informative priors for key population processes—immigration, cub recruitment and non-culling mortality–that could not be quantified in the field. Using simulated datasets we showed that the model reliably estimated fox density and demographic parameters, and we showed that conclusions drawn from real data were robust to alternative model assumptions. All estates achieved suppression of the fox population, with pre-breeding fox density on average 47% (range 20%–90%) of estimated carrying capacity. As expected, the number of foxes killed was a poor indicator of effectiveness. Estimated rates of immigration were variable among estates, but in most cases indicated rapid replacement of culled foxes so that intensive culling efforts were required to maintain low fox densities. Due to this short-term impact, control effort focussed on the spring and summer period may be essential to achieve management goals for prey species. During the critical March-July breeding period, mean fox densities on all estates were suppressed below carrying capacity, and some maintained consistently low fox densities throughout this period. A similar model will be useful in other situations to quantify the effectiveness of lethal control on restricted areas.

**Funding:** The work was funded entirely by the Game & Wildlife Conservation Trust (www.gwct. org.uk), registered as a charity in England & Wales (1112023) and Scotland (SC038868).

**Competing interests:** The authors have declared that no competing interests exist.

## Introduction

Lethal control of predator numbers to benefit prey species is controversial and there is increasing demand for objective analysis of its benefits and costs [1,2]. Benefits may be evidenced by showing increased prey survival or breeding success where predators are culled, but because of variation in these parameters by site and year, unequivocal demonstration of the effect of predator control requires a costly experimental approach (e.g. [3,4]). Because of their cost and difficulty, such field experiments are rare, and do not sample a range of situations or operators. Hence, they are limited in both generality and relevance to any different situation. The fact that predator culling benefits prey in one context does not imply that it will do so in another. Predator-removal experiments generally avoid the use of research techniques that might interfere with the outcome, and consequently provide little detail on the human-predator-prey interactions that contribute to their success or failure. One of the components likely to vary among different situations with different operators is the impact of predator culling on local predator density [5]. A common perception is that culled predators are rapidly replaced through immigration, and this is used to argue both the futility of predator control, and its difficulty and importance [6–9]. With few exceptions (e.g. mink [10]) almost nothing is known of the dynamics of predator numbers during control efforts.

In rural Britain, red foxes (*Vulpes vulpes*) are commonly culled within the boundaries of game-shooting estates or wildlife reserves, with the aim of suppressing fox density locally (i.e. within the estate or reserve [11]). Such management units are typically <50 km$^2$ in area. In England and Wales, fox control of some kind takes place on 43% of agricultural landholdings larger than 5 ha, but is particularly associated with larger landholdings (i.e. the 22% > 100 ha), a game-shooting interest, and arable or upland land-classes [12]. At larger, regional scales (>1,000 km$^2$) there is considerable variation in fox density and demographics attributable to regional differences in food availability and fox culling intensity [13–16]. Consequently, the relationship between any estate and its geographical context will influence within-estate fox population dynamics. Cull data from individual estates suggest that the rate at which culled foxes are replaced by immigration may vary greatly among estates due to local circumstances [17]. At the same time, the within-estate environment (e.g. habitats, topography) as well as operator skills, resources, effort and strategy will determine which culling methods are employed and how effective they are [18]. Understanding the impact of culling on fox density within individual estates would allow adaptive management at that location, and also lead to more general conclusions.

Most established field methods to estimate population density and demographic parameters (e.g. mark-recapture), have limitations when applied at the local scale, or in the rapidly changing situation expected where culling is intensive and replacement of culled animals through immigration is rapid. In these circumstances, index methods (e.g. camera traps, faecal surveys) are arguably most appropriate to track changes in local abundance ([19] but see [20]). Given suitable survey design and demanding assumptions these can be used to estimate density (e.g. [21]). But culling actions themselves create population perturbations that are potentially informative. Combined with a suitable abundance index, cull data can be used to fit a population dynamics model to reconstruct within-year population density. The model parameter values most likely to have generated the data (i.e. giving the closest fit of the model to the data) are taken as estimates of the population parameters according to both likelihood and Bayesian statistical paradigms [22].

Previous modelling of culled fox populations on restricted areas has shown that immigration was important on those sites studied. Lieury et al. [7] sought to explain observed removals from culling by fitting models accounting for immigration to fox densities estimated from

costly site-specific surveys. Harding et al. [23] and McLeod and Saunders [24] used parameter values from the literature to reconstruct fox populations with density similar to those observed, then inferred the rate of replacement which explained the known removals. Because parameter values are likely to vary substantially between sites, conclusions from the fixed population reconstruction approach are necessarily less reliable than if site-specific parameter estimates were obtained.

State-space models [25,26] are a type of hierarchical model that can account for uncertainty in parameter values, process variation (environmental and demographic stochasticity), and observation error in non-linear, non-normal systems with density-dependence [27,28]. A state-space model contains a state-process sub-model which specifies the probability distributions of deterministic parameters associated with population processes; these combine to predict latent (unobserved) states, e.g. population density, which are subject to process variation. A corresponding observation error sub-model defines the probability distribution of the observations, which are related to the latent states by parameters that define a relationship between them [29]. Bayesian algorithms such as Markov chain Monte Carlo (MCMC) may be used to find the joint posterior parameter distribution, with previous knowledge on parameters incorporated using prior probability distributions.

In the present paper we aimed to reconstruct changes in fox density on individual game-shooting estates in Britain during the culling process, which was continuous in most cases. We did this by fitting a novel population dynamics model to data on culling effort and success using a Bayesian state-space modelling framework to quantify the major sources of uncertainty. The results allow us to determine the effect of culling on within-estate fox numbers and to evaluate the relationships between culling effort, foxes culled, and fox density in restricted-area culling.

## Methods

### Data

We used data collected by the Game & Wildlife Conservation Trust (GWCT) during January 1996 to August 2000 from a self-selecting sample of 74 estates where gamekeepers culled foxes primarily by 'lamping' (i.e. using a spotlight and rifle to find and shoot foxes at night) and were willing to keep daily records in a pro-forma diary of lamping effort; foxes seen; and foxes culled (categorised by culling method). These data allowed calculation of lamping detection rate for use as an abundance index. Based on a preliminary assessment of data requirements to fit the model using simulation-estimation analysis, we used data from a sub-sample of 22 estates that contributed data for at least three successive years (S1 Table). Estates ranged in size from 1.6 to 36.4 km$^2$ (mean = 8.4 km$^2$; S1 Table) and most were located in arable or lowland pastural landscapes (Fig 1). In this paper, estates are identified by random three letter codes to preserve confidentiality.

Given the preferential use of lamping by contributing gamekeepers, accounting for a mean of 66% of the adult foxes killed across the 22 estates, annual lamping effort per km$^2$ provided an index of the intensity of fox culling effort. This varied considerably between estates and over time (S1 Table). Information on the sex of foxes killed was incomplete, and there was no information on the age of culled foxes, apart from the numbers of cubs killed at breeding earths. The intensity of fox control on neighbouring areas was unknown.

A Bayesian state-space population dynamics model contains latent states for each time step in addition to deterministic parameters. This required us to aggregate the data. Modelling data on too short a time step (e.g. daily) would result in an over-parameterised model that would exhibit prohibitively slow convergence [26]; using too long a time step (e.g. annual) would lose

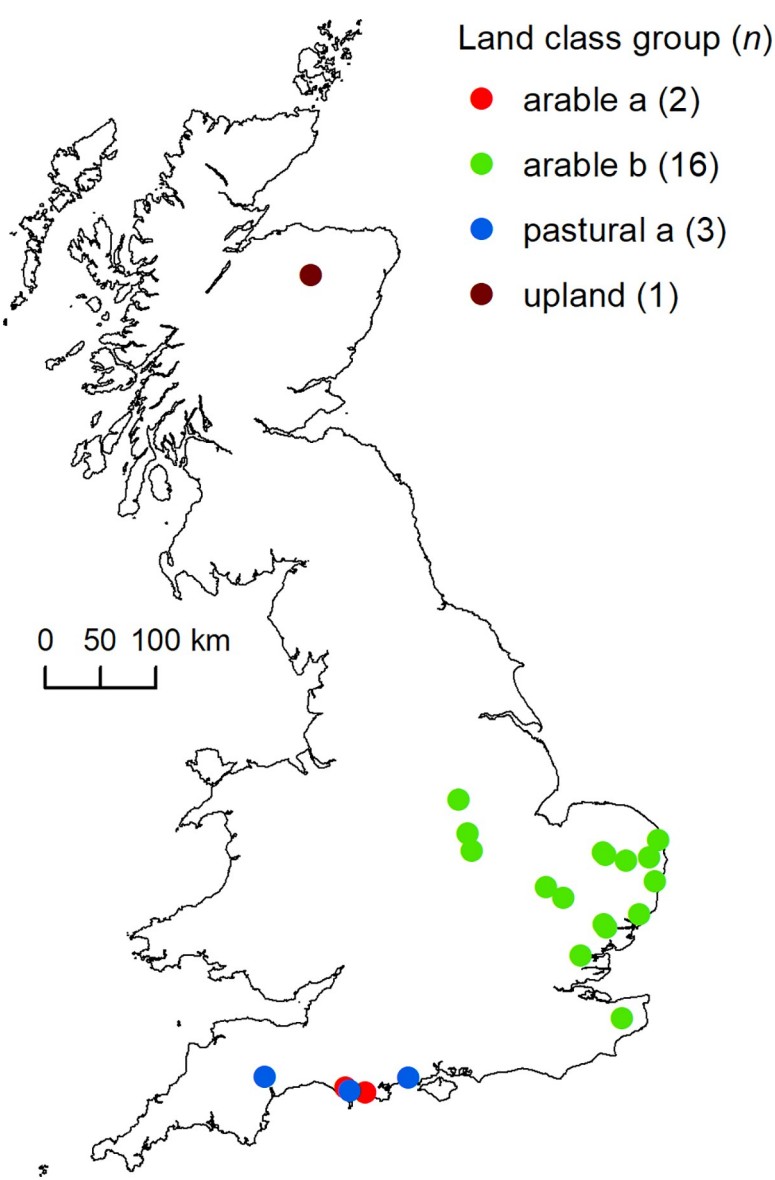

**Fig 1. Locations of the 22 modelled estates showing the landscape type each estate was within.** Landscape types were determined by groupings of land class strata which summarise variation in ecological, physio-geographical, and human geographical attributes (see [30] for details).

information about the effect of removals on detection rate that could allow immigration and reproduction to be disentangled during within-year population dynamics. Compared to environmental processes, demographic processes that affect culled fox populations, such as immigration, may operate on relatively short time-scales. While telemetry data show that replacement of territory-holding foxes removed by culling can occur after only one week, movement can occur over even shorter time scales, e.g. if a vixen moves a litter of cubs within her territory but over an estate boundary [31]. In addition, from the management perspective the average gamekeeper goes lamping on one night per week in response to fox detections (unpublished data, GWCT).

To balance the above factors, weekly or two-weekly aggregation of daily data seemed suitable. Even when aggregated, these datasets were large, with >100 time-steps for some estates. From initial model runs, posterior median fox density and other model parameter estimates from a weekly time-step model differed from those on a two-weekly time-step by <±5%, so to minimise computation time we chose to aggregate data into a two-weekly time-step. Cull data were converted to density of foxes killed per km$^2$ using each estate area (S1 Table). For each two-weekly time-step $t$, the data thus consisted of the number of lamping hours $E_t$, the observed number of fox detections $Y_t$, the density of foxes culled by lamping $L_t$, the density of foxes culled by snaring/other methods $S_t$, and the density of fox cubs culled at earths $C_t$ (Table 1, S1 Dataset).

## Population process model

The lack of age- or sex-structured information in the culling data limited our choice of population dynamics modelling approach, i.e. matrix models were unsuitable. Instead we used a form of depletion model [32–34]. Depletion models infer population density by relating observed changes in an abundance index to the known scale of deliberate removals by harvest or culling [34]. Depletion models are well-suited to a Bayesian state-space framework as they are similarly structured with population and observation sub-models, and are designed to capture within-year dynamics [35]. A limitation of classic depletion models is that they include no mechanism for a local population to recover if all individuals are removed, i.e., they assume closed populations, but these models can be generalised to allow immigration to contribute to population density [34].

Use of lamping detection rate as the abundance index required that we model the subset of the fox population vulnerable to detection by gamekeepers. The 'observable' population density on a given estate in time-step $t$, $N_t$, comprises adult foxes and cubs that are active away

**Table 1. Symbols and description of data variables and model parameters.**

| Notation | Description |
| --- | --- |
| *Subscripts* | |
| $t$ | Time, two-weekly time-step |
| *Model parameters* | |
| $N_0$ | Fox density in week 0, fox km$^{-2}$ |
| $K$ | Fox density at carrying capacity, fox km$^{-2}$ |
| $v$ | Immigration rate, fox km$^{-2}$ 2wk$^{-1}$ |
| $r$ | *Per capita* birth rate, cub fox$^{-1}$ yr$^{-1}$ |
| $M$ | Instantaneous non-culling mortality rate, 2wk$^{-1}$ |
| $d$ | Rate of successful search, km$^2$ hr$^{-1}$ |
| $\sigma_p$ | Standard deviation in process error (time-step specific) |
| *States* | |
| $N_t$ | Fox density excluding cubs prior to weaning, fox km$^{-2}$ |
| $\varepsilon_t$ | Process error |
| *Data variables* | |
| $w_t$ | Proportion of cubs reaching weaning age |
| $E_t$ | Hours of lamping effort |
| $Y_t$ | Number of foxes sighted |
| $L_t$ | Lamping cull, fox km$^{-2}$ |
| $S_t$ | Snaring (and other method) cull, fox km$^{-2}$ |
| $C_t$ | Cub cull at earths, cub km$^{-2}$ |

from the breeding earth following weaning. In Britain, most cubs are born between mid-March and mid-April [36]. Fox cubs are fully weaned at 6–8 weeks of age [36] and become active away from the earth by eight weeks [37]. Rather than assume all cubs recruited in one time-step, we determined a distribution by which cubs would recruit to the observable population post-weaning. To allow for variation in timing of breeding events within and between estates, we used data from fox populations in south-east England and Wales describing the probable conception dates of female foxes killed during pregnancy [37]. We fitted a logistic distribution to these data to determine the distribution of conception events in the female fox population over time and inferred a schedule $w_t$ of weaning events based upon timing of birth and weaning relative to conception (see S1 Appendix for details).

We assumed a 50:50 sex ratio with all females breeding, and that productivity was similar across ages. Given these assumptions, we modelled recruitment of weaned cubs into the observable population of both sexes using a *per capita* birth rate parameter, *r*. The proportion of weaned cubs recruiting to the observable population during each time-step was determined by schedule $w_t$ (S1 Appendix: Fig A, *sensu* [38]). Cubs culled at earths were assumed to be of pre-weaning age. To account for cubs being killed at earths at any time from birth until weaning age, rather than just in the time-step they recruit, all cubs culled at earths in year *y* were summed for each estate and re-distributed into each recruitment time-step *t* following the $w_t$ schedule to calculate normalised values for $C_t$ within each year:

$$C_{t,y} = w_t \sum_{y=1996}^{2000} C_{t,y}^{obs} \tag{1}$$

This ensured that cubs culled at earths within each year were not removed from the model before they had been produced.

Culling was expected to be a substantial additive component of overall fox mortality [13], but local fox density will also be determined by non-culling mortality factors including natural risks, e.g. from disease or starvation, and non-natural risks, e.g. from road traffic collisions or secondary poisoning [15,36]. Epidemic diseases, e.g. sarcoptic mange, were not a feature of the rural fox population at the time of data collection. Other factors, such as road traffic collisions, are expected to be density-independent. We therefore assumed that non-culling mortality was a density-independent constant risk which we modelled using an instantaneous non-culling mortality rate, *M*.

We modelled net immigration using a parameter for the immigration rate of foxes per km$^2$ per time-step, *v*. Both recruitment of weaned cubs and net immigration into the observable fox population were assumed to be density-dependent processes, i.e. related to fox density within the population modelled [7,13,39]. By adding cubs into the population post-weaning following the $w_t$ schedule, non-culling mortality of pre-weaned cubs is implicitly incorporated in *r*. In line with other authors [24,38,39], density-dependence in seasonal reproduction and/or immigration was modelled using logistic terms with a carrying capacity, *K*, describing the fox density above which no cubs are recruited and there is no immigration into the population. In absence of suitable covariate information, e.g. on seasonal variation in food resources, *K* was assumed to be constant on each estate during the period covered by the data.

The model is conditioned on the cull between *t-1* and *t* and is initialised by $N_0$, the initial fox density at the start of the detection rate time series. Based upon the above schedule and parameter definitions (units described in Table 1), $N_t$ was assumed to follow the state

equations:

$$N_1 = N_0 e^{\varepsilon_1 - (\sigma_p^2/2)} \tag{2}$$

$$N_t = [N_{t-1} e^{-M} + \nu(1 - N_{t-1}/K) + J_{t-1} - L_{t-1} - S_{t-1}] e^{\varepsilon_t - (\sigma_p^2/2)} \tag{3}$$

$$J_t = w_t r N_t (1 - N_t/K) - C_t \tag{4}$$

where $J_t$ represents the weaned cubs recruiting during $t$. State-space models can allow for implausible predictions, such as negative values of density [40]. In a model conditioned on the observed cull, the risk of negative predictions for density was further increased in time-steps with relatively large removals, so for numerical stability we constrained $N_t$ within the model to be positive ($\geq$0.001 fox km$^{-2}$).

Process error aims to capture any additional variation in population trends due to factors not included in the model [41]. Lognormal process errors $\varepsilon_t$ were obtained by sampling from a standard normal distribution with standard deviation $\sigma_p$:

$$\varepsilon_t \sim N(0, \sigma_p) \tag{5}$$

The lognormal bias correction factor ($-\sigma_p^2/2$) was applied to the state equations (Eqs 2 and 3) to reflect expected $N_t$ rather than median $N_t$.

Estimation of both process and observation uncertainties using state-space models is challenging [42], and initial modelling that estimated $\sigma_p$ using a vague prior suggested that it was only weakly identifiable. This had potential to influence the reliability of other parameter estimates, e.g. carrying capacity, due to confounding between the strength of density-dependence and process error [28]. To overcome this problem, we chose to fix $\sigma_p$ at a value of 0.2 based on knowledge of fox ecology and the wider literature on state-space modelling applications (S2 Appendix: Table A), and to consider the consequences of our choice through sensitivity analysis.

## Observation model

For each estate, the model was fitted to the observed number of fox detections, $Y_t$, with the observation errors assumed to be Poisson distributed:

$$\hat{Y}_t = d E_t N_t \tag{6}$$

$$Y_t \sim \text{Poisson}(\hat{Y}_t) \tag{7}$$

where $d$ is the rate of successful search in units km$^2$ hr$^{-1}$, assumed to be constant over time, and $E_t$ is the number of lamping hours in $t$. The equation for the expected number of detections $\hat{Y}_t$, Eq 6, is the Holling disc equation [43] with a handling time of zero. As fox densities are relatively low, this assumption is reasonable in populations undergoing culling by gamekeepers, with minimal effects on density estimates of incorporating unrealistically large handling times [44]. However, incorporating handling time into Eq 6 may be necessary where fox densities are much greater than those encountered in Britain.

Incorrect specification of the observation model can markedly affect inference using state-space models [45]. The Poisson distribution cannot account for any over-dispersion in the number of detections caused by time-steps in which there was no lamping effort. Overdispersion may also be caused by variation in weather, moonlight, habitat, or operator experience. We therefore explored use of the negative binomial distribution to model observation error in

the detection data, but found the results were not sensitive to this choice (so we do not report them here).

## Prior probability distributions for parameters

We used informative priors for $v$, $M$, $r$, and $d$ (Table 2). The prior for immigration rate was obtained from an analysis of fox culling data from GWCT's National Gamebag Census [17]. This gave a lognormally-distributed prediction for $v$ across all landscape types with a median of 2.41 fox km$^{-2}$yr$^{-1}$ and a coefficient of variation (CV) of 0.84. The prior for instantaneous non-culling mortality rate was obtained from a meta-analytic model that predicts annual $M$ across varied taxonomic groups from maximum age data, which for fox populations in Britain was assumed to be nine years, giving a lognormal distribution with median of 0.34 yr$^{-1}$ and a CV of 0.58 [46]. This is equivalent to a finite annual mortality rate (or the proportion of the population that suffers non-culling mortality in a given year, obtained as $1-e^{-M}$) of 0.29. For use with data on a two-weekly time-step, $v$ and $M$ were transformed from annual rates to two-weekly rates by dividing by the 26 fortnights per year. For easier interpretation, these two parameters are presented in the results as weekly rates.

The prior for *per capita* birth rate was developed from an analysis of data on litter size per female, giving a gamma-distributed mean $r$ prediction of 3.17 cub fox$^{-1}$yr$^{-1}$ ± 1.07 s.d. (S3 Appendix). We obtained a prior for the rate of successful search using a model that used a mechanistic understanding of the gamekeeper lamping search process, combined with empirical data from a distance sampling survey of foxes using similar methodology [44]. This gave a lognormally-distributed prediction for $d$ with a median of 2.01 km$^2$hr$^{-1}$ and a CV of 0.56.

For the remaining estimated parameters in the model, $N_0$ and $K$, we used vague priors which took the same uniform distribution with the lower bound at 0.001 fox km$^{-2}$ and the upper bound equal to the fox density in an urban population from Bristol (13.9 fox km$^{-2}$ [47]). The Bristol population was at this density for several years prior to a period when the density tripled and an epizootic of mange occurred, so it was considered to be a 'healthy' urban density representing the maximum a rural population might be expected to reach if food and den site availability in rural areas were increased to urban levels, and there were no sympatric carnivores, e.g. badgers, that could affect fox density [48]. As there was no knowledge of the culling history on each estate, $N_0$ was assumed able to be any value between 0 and $K$, but not higher.

**Table 2. Prior probability distributions for estimated model parameters.**

| Model parameter | Units | Distribution | Parameters | Reference |
|---|---|---|---|---|
| $N_0$ | fox km$^{-2}$ | ~ uniform | lower = 0.001 | [47] |
| | | | upper = 13.9 | |
| $K$ | fox km$^{-2}$ | ~ uniform | lower = 0.001 | [47] |
| | | | upper = 13.9 | |
| $v$ | fox km$^{-2}$2wk$^{-1}$ | ~ lognormal | median = ln(2.41/26) | [17] |
| | | | CV = 0.84 | |
| $r$ | cub fox$^{-1}$yr$^{-1}$ | ~ gamma | $a$ = 8.77 | S3 Appendix |
| | | | $b$ = 2.76 | |
| $M$ | 2wk$^{-1}$ | ~ lognormal | median = ln(0.34/26) | [46] |
| | | | CV = 0.58 | |
| $d$ | km$^2$hr$^{-1}$ | ~ lognormal | median = ln(2.01) | [44] |
| | | | CV = 0.56 | |

## MCMC simulations

Samples from the joint posterior probability distribution of the unknown parameters and latent states $p(N_0, K, v, r, M, d, N_t \mid data)$ were simulated by MCMC integration using Win-BUGS 1.4 [49] implemented from within the R statistical software [50] using the R2WinBUGS package [51]. To reduce correlation between $N_t$ and $K$, and thereby improve the slow mixing of the Gibbs sampler over the support of the joint posterior, the model was re-parameterised by expressing fox density as a proportion of carrying capacity ($P_t = N_t / K$ [52]). The state equations were rewritten as:

$$P_1 = (N_0/K)e^{\varepsilon_1 - (\sigma_p^2/2)} \tag{8}$$

$$P_t = [P_{t-1}e^{-M} + v/K(1 - P_{t-1}) + G_{t-1} - L_{t-1}/K - S_{t-1}/K]e^{\varepsilon_t - (\sigma_p^2/2)} \tag{9}$$

$$G_t = w_t r P_t(1 - P_t) - C_t/K \tag{10}$$

where $G_t$ is the re-parameterisation of $J_t$. The observation equation was rewritten as:

$$Y_t \sim Pois(dE_t P_t K) \tag{11}$$

For presentation of fox density in the results, we obtained the marginal posterior estimates of $N_t$ as the product of $P_t$ and $K$. BUGS code is described in S1 File.

The joint posterior was estimated from two independent MCMC chains run in parallel with initial values chosen randomly from the joint prior. To conserve computer memory, only 1 in 100 iterations of the Markov chains were recorded after the first 100,000 iterations were removed as the burn-in. Inferences were derived from a sample of 20,000 iterations from two chains of 10,000 iterations. Convergence of the Markov chains to the posterior distribution was diagnosed by visual inspection of parameter trace plots and use of the R coda package [53]. Gelman-Rubin convergence statistics [54] were <1.01 for all parameters and states, suggesting the chains had fully converged.

Post-model-pre-data probability distributions are a diagnostic showing how the priors interact with a model given the culling data but before the model is fitted to an abundance index [55]. Estimating post-model-pre-data distributions does not involve updating of the priors as sampling is performed directly from the joint prior probability distribution. This allows evaluation of the extent to which fitting the model to detection rate data updates the distributions determined by the interaction of the priors and inputted culling data within the model formulation. We estimated the post-model-pre-data distribution for fox density in the final time-step and later compared it to the posterior for fox density in the final time-step to show the extent of posterior updating on each estate.

## Culling vs. non-culling mortality

The relative contributions to fox mortality from culling and from non-culling factors were assessed by using the estimates of $N_t$ and $M$ to calculate non-culling mortality $X_t$ as:

$$X_t = \tilde{N}_t - \tilde{N}_t e^{-\tilde{M}} \tag{12}$$

where $\tilde{N}_t$ is the posterior median fox density and $\tilde{M}$ is the posterior median for instantaneous non-culling mortality rate on a two-weekly time-step. Culling mortality $T_t$ was calculated as:

$$T_t = L_t + S_t \tag{13}$$

The cumulative mortality (expressed as dead foxes km$^{-2}$) from both culling and non-culling

mortality was calculated on each estate and examined relative to estimated fox density and carrying capacity.

## Simulation-estimation analysis

A non-identifiable model has redundant parameters; this may either be intrinsic redundancy, which is a model property, or extrinsic redundancy, meaning not all model parameters are identifiable for a specific dataset [56,57]. When generalising a depletion model for open populations, the risk of non-identifiability is increased due to addition of parameters to account for density-dependent cub recruitment and immigration, so sources of estimation error and bias should be examined to determine whether the available data contain sufficient information to identify all parameters [34]. Insufficient information in the data can also lead to parameter identifiability problems in data-hungry state-space models [27,57,58], and ecologists using state-space models are urged to assess whether the parameters can be reliably estimated before drawing conclusions from their results [59]. Model performance can be examined using simulation-estimation analysis, in which models are fitted to simulated data where the true values of the parameters are known. Simulation-estimation analysis can help understand what features make data informative or uninformative and enable identification of datasets for which a model is likely to produce biased or imprecise estimates. It has previously been used in fisheries science to assess depletion model performance [35,60].

We performed simulation-estimation analysis to examine the performance of our generalised depletion model, with the aim of understanding the reliability of latent state and parameter estimates when using 1) vague or informative priors on model parameters, and 2) different time series lengths. We describe these analyses in detail in S4 Appendix, but outline the steps involved here. First, we generated lamping effort data over different lengths of time to drive simulation of detection and culling time series data. Second, using the above population process and observation models we simulated the dynamics of sets of 20 known ('true') observable fox populations subjected to culling using an 'operating model' with known parameter values, process errors, and observation errors [61,62]. Third, we used the above Bayesian estimation methods (with either vague or informative priors) to obtain the joint posterior probability distribution of latent states and parameters from the simulated data for each population on numbers of foxes detected and culled in each time step. Last, we measured estimation bias by comparing the posterior median values with the known fox density and parameter values.

## Sensitivity analysis

The sensitivity of the model parameter and fox density estimates to different types of assumption was examined as follows. The analysis was performed on a sample of six estates that represented estates covering a range of sizes and culling intensities.

**1. Structural assumptions.** In the population model, immigration was assumed to be a year-round constant process. We examined the alternative assumption that immigration was a seasonal process around the August to March dispersal period [63] by running a version of the model where no immigration occurred during April-July. This was achieved by populating a one-year long binary vector, i.e. 26 two-week time-steps, with a zero if the time-step was in April-July (4 months) and a one if the time-step was in August-March (8 months). To maintain the same annual immigration rate for comparison between assumptions, we scaled the 'ones' upwards to ensure the sum of this vector totalled 26. To increase immigration rate only during the eight-month dispersal period, the vector was then used to multiply $v$ in Eq 3 in each time-step. The results from the reference model assuming constant immigration were then compared to those from the model assuming seasonal immigration. We also examined the

timing of cub recruitment. This was achieved by moving the weaned cub distribution two weeks earlier or two weeks later to account for cubs being born on average earlier or later across the estate.

**2. Prior probability distribution specification.** For the upper bound of the vague uniform prior for $K$, we used the spring density (13.9 fox/km$^2$) in a British urban population during a relatively stable pre-mange period [47]. In the spring preceding a mange epizootic, this population reached 25.8 fox km$^{-2}$, and then 37.0 fox km$^{-2}$ in the first year of the ensuing epizootic period. We examined sensitivity to a higher upper bound for $K$ of 25.8 fox km$^{-2}$, as a rural population could in theory reach this high density if there was enough food available. We also examined sensitivity to a lower upper bound for $N_0$ of 6.95 fox km$^{-2}$. The informative lognormal prior chosen for $M$ was based upon a model prediction of $M$ from maximum age, as this allows for both extrinsic and intrinsic mortality factors to affect predicted mortality [46]. An alternative meta-analytic model can predict $M$ from body mass, but this model only allows for intrinsic mortality factors to affect predicted mortality. We examined sensitivity of the results to our choice of prior for $M$ using a lognormal prior based upon the body mass prediction of $M$, where the prior median was $\ln(0.27/26)$ with a CV of 0.60 [46]. The informative prior for $r$ developed in S3 Appendix had a lower CV (0.34) than the >0.5 value recommended for the CV of informative priors in population dynamics models [64], so we examined sensitivity of the results to this prior by using a vague uniform prior for $r$ with a lower bound at zero and an upper bound at 6.0 cub fox$^{-1}$yr$^{-1}$.

**3. Process error specification.** The decision to fix $\sigma_p$ at 0.2 was examined by 1) use of alternative fixed values 0.05 and 0.1; 2) use of priors that made $\sigma_p$ estimable; and 3) calculation of empirical $\sigma_p$. Two priors were assessed 1) a vague uniform prior with a lower bound at 0.001 and an upper bound at 1.0, and 2) an informative lognormal prior with a median equal to $\ln(0.05)$ and a CV equal to 0.2, which gave high certainty around a small value for $\sigma_p$. Empirical $\sigma_p$ provides an indication of whether the realised process errors $\varepsilon_t$ from the estimation are smaller or larger on average than assumed in the model. The empirical estimates of $\sigma_p$ are dependent upon the structural assumption for the observation errors. We extracted the vector of $\varepsilon_t$ that is the length of the time series for each MCMC iteration as the difference between the deterministic parts of Eq 8 and Eq 9 that are not subject to lognormal process error, i.e. those parts between parentheses/brackets, and the predicted value of $P_t$ that is subject to process error. Empirical $\sigma_p$ is then calculated as the standard deviation in $\varepsilon_t$. These values can then be summarised for each estate by taking the mean over the number of iterations.

## Results

### Simulation-estimation analysis

We summarise results from the simulation-estimation analyses here (described in detail in S4 Appendix) to maintain the focus of this paper. When informative priors were used on model parameters, population density was reliably reconstructed from detection and cull data. Mean percent relative bias in posterior median density estimates compared to true density was only +3% across 20 simulated culled populations, compared to +10% when vague priors were used (S4 Appendix: Figure B). There was strong bias in some model parameter estimates when using vague priors, particularly for $N_0$, $K$, and $M$, but this was reduced when informative priors were used (S4 Appendix: Figure C). In each simulated population, the true values for parameters were found within 80% credible intervals when either vague or informative priors were used for all parameters, with the exception of $N_0$ where in 4 of 20 populations the true value was included only within the 95% credible

interval (S4 Appendix: Figure D). As already described, conclusions from the simulation-estimation analysis were used to sub-sample those 22 out of 74 estates that met the data requirements elaborated below.

As would be expected, reliable estimation of $r$ was possible only when sufficient detection rate data were recorded during the cub recruitment period. Even when using an informative prior on $r$, estimates were worse in datasets where the number of weeks with no detection data in May/June was greater. Moderate negative posterior correlation between $r$ and $v$ was noticeable in these datasets, highlighting that there was insufficient information in the data to disentangle these processes. This led to a data requirement for at least two timesteps containing lamping effort during the cub recruitment period. This is typically the most challenging time of year in which to detect foxes with a lamp on arable land due to increased crop cover, and consequently some estates did not lamp at all during this period. This led to their exclusion from our modelling study.

Bias in model parameter estimates also depended upon the length of data time series. When the data covered fewer than three cub recruitment periods, estimates of some parameters, i.e. $N_0$, $K$, were strongly positively-biased and this resulted in overestimation of density (S4 Appendix: Figure F). In shorter time series, correlation between $r$ and $v$ became a problem, so we determined that a further data requirement was for $\geq 3$ cub recruitment periods to enable reliable estimation of both parameters. Additional data requirements were for detection rate time series to contain a minimum of eight weeks of lamping effort per year, with no gap in the time series longer than nine months. Time series with little lamping effort in the first year resulted in greater $N_0$ estimation bias; consistent lamping effort during the first three months was thus also a requirement for reliable estimation of $N_0$.

## Parameter estimation

Detection and effort data were modelled on a two-weekly time-step to estimate model parameters on the 22 estates. The priors for most model parameters were updated on all estates (Fig 2). The greatest marginal posterior updates were for those parameters which had vague priors, $N_0$ and $K$. Posterior medians for $N_0$ ranged between 0.24 and 6.76 fox km$^{-2}$, with most estates being below 3.5 fox km$^{-2}$ (Table 3). CV for $N_0$ was generally greater than 0.5, with the estimates with larger CV characterised by the detection rate time series on these estates being more variable during the first few weeks of data. Posterior medians for $K$ ranged between 1.93 and 8.25 fox km$^{-2}$.

Parameters with informative priors were also updated, particularly $v$ and $d$. Posterior medians for $v$ ranged between 0.023 and 0.564 fox km$^{-2}$ wk$^{-1}$. Most of the updates were an increase relative to the prior median of 0.046 fox km$^{-2}$ wk$^{-1}$. The highest estimate for $v$ was from estate YZM which had the greatest culling effort intensity and annual bag density. Posterior medians for $d$ ranged from 0.19 to 0.85 km$^2$ hr$^{-1}$, which on all estates meant that the updates were a decrease relative to the prior median of 2.01 km$^2$ hr$^{-1}$. Posterior medians for $r$ ranged between 1.18 and 3.69 cub fox$^{-1}$yr$^{-1}$. The amount of updating was variable and for many estates the posterior median was similar to the prior median of 3.06 cub fox$^{-1}$yr$^{-1}$, with the posterior precision being only a small improvement on the prior precision (CV = 0.34). $M$ was the parameter with least updates to the prior median of 0.0065 wk$^{-1}$, as the posterior medians ranged between 0.0055 and 0.0073 wk$^{-1}$. The precision was usually not an improvement on the prior precision (CV = 0.58).

Across all estates, posterior correlations among parameters were not strong as correlation coefficients were all lower than 0.7. There were moderate posterior correlations (0.4–0.7) between $N_0$ and $d$ (negative) on 17/22 estates, between $K$ and $d$ (negative) on 11/22 estates,

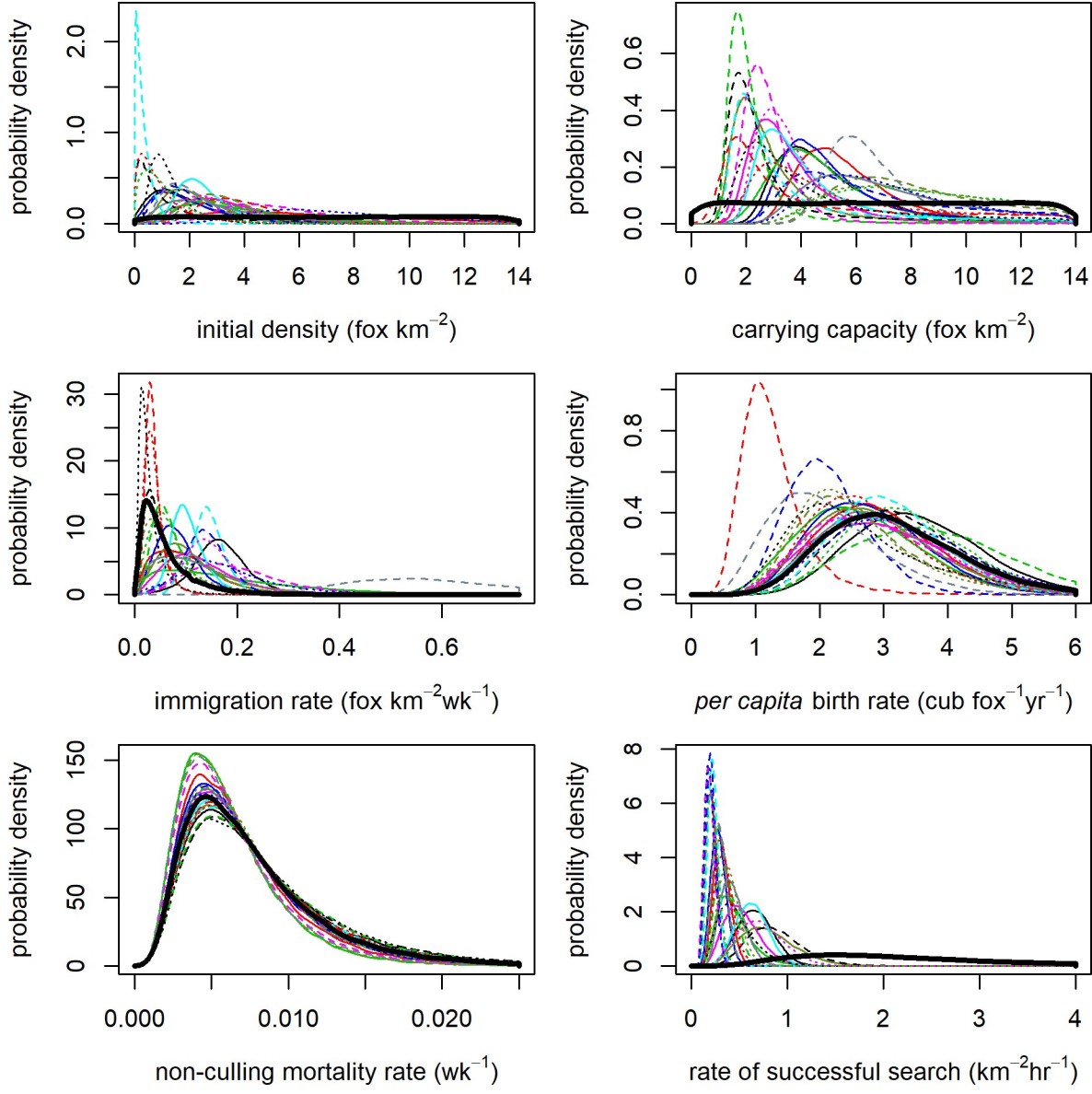

**Fig 2. Marginal posterior probability distributions for the estimated parameters from the 22 estates:** $N_0$ **(initial density),** $K$ **(carrying capacity),** $v$ **(immigration rate),** $r$ **(***per capita*** birth rate),** $M$ **(non-culling mortality rate), and** $d$ **(rate of successful search).** Each estate is represented by the same line type and colour combination in all panels. The prior probability distribution for each parameter is shown by the bold black line. Differences between posteriors and priors indicate updating of parameter estimates through fitting the model to the data.

and between $K$ and $v$ on 6/22 estates. Posterior correlation coefficients between all other parameters reflected weak posterior correlation.

## Density reconstruction

We modelled fox population density to allow comparison between estates. The model fitted the observed detection rate time series well on all estates despite large differences between estates in detection rates (Fig 3A, S5 Appendix: Figures A(a)-U(a)), with a maximum detection rate of 0.81 fox seen hr[-1] on HIR compared to a maximum of >5 fox seen hr[-1] on CHU, EWE,

**Table 3. Marginal posterior median (and CV) for estimated parameters on each estate.**

| Estate | $N_0$ | $K$ | $v$ | $r$ | $M$ | $d$ |
|---|---|---|---|---|---|---|
| | fox km$^{-2}$ | fox km$^{-2}$ | fox km$^{-2}$wk$^{-1}$ | cub fox$^{-1}$yr$^{-1}$ | wk$^{-1}$ | km$^2$ hr$^{-1}$ |
| BMM | 2.67 (0.67) | 6.57 (0.37) | 0.106 (0.68) | 2.96 (0.33) | 0.0063 (0.60) | 0.42 (0.30) |
| CHU | 5.61 (0.52) | 5.39 (0.34) | 0.093 (0.66) | 2.85 (0.32) | 0.0060 (0.58) | 0.31 (0.27) |
| CIP | 0.72 (1.06) | 4.44 (0.58) | 0.035 (0.56) | 2.62 (0.33) | 0.0069 (0.63) | 0.33 (0.29) |
| CUL | 3.27 (0.52) | 3.27 (0.48) | 0.123 (0.72) | 2.98 (0.37) | 0.0064 (0.61) | 0.51 (0.35) |
| DLQ | 6.76 (0.37) | 6.76 (0.35) | 0.168 (0.66) | 3.19 (0.33) | 0.0064 (0.61) | 0.19 (0.28) |
| DWS | 1.65 (0.79) | 4.61 (0.42) | 0.167 (0.33) | 3.47 (0.29) | 0.0070 (0.67) | 0.66 (0.28) |
| EWE | 2.95 (0.53) | 3.38 (0.42) | 0.144 (0.40) | 2.82 (0.35) | 0.0064 (0.61) | 0.71 (0.33) |
| FAH | 0.82 (1.36) | 2.18 (0.82) | 0.048 (0.81) | 3.20 (0.31) | 0.0073 (0.66) | 0.85 (0.34) |
| FHC | 5.03 (0.40) | 2.63 (0.37) | 0.189 (0.50) | 2.79 (0.38) | 0.0057 (0.58) | 0.19 (0.29) |
| GDE | 2.17 (0.60) | 2.50 (0.63) | 0.098 (0.62) | 2.77 (0.34) | 0.0068 (0.63) | 0.77 (0.36) |
| GHT | 1.82 (0.63) | 5.94 (0.43) | 0.137 (0.33) | 2.06 (0.30) | 0.0063 (0.60) | 0.20 (0.25) |
| HIR | 0.24 (1.01) | 2.53 (0.75) | 0.142 (0.24) | 3.03 (0.27) | 0.0069 (0.66) | 0.22 (0.23) |
| HUS | 3.41 (0.57) | 1.93 (0.63) | 0.065 (0.60) | 3.69 (0.32) | 0.0072 (0.66) | 0.35 (0.31) |
| LEL | 1.14 (0.61) | 3.39 (0.66) | 0.023 (1.01) | 2.42 (0.44) | 0.0073 (0.65) | 0.44 (0.42) |
| MAH | 2.38 (0.40) | 3.71 (0.53) | 0.101 (0.36) | 2.99 (0.34) | 0.0067 (0.63) | 0.63 (0.27) |
| NOG | 1.88 (0.84) | 4.72 (0.41) | 0.081 (0.52) | 2.72 (0.32) | 0.0062 (0.59) | 0.28 (0.27) |
| NYP | 3.29 (0.62) | 4.46 (0.41) | 0.157 (0.76) | 2.56 (0.36) | 0.0055 (0.55) | 0.42 (0.41) |
| OCS | 0.94 (0.92) | 8.25 (0.32) | 0.060 (0.60) | 3.30 (0.29) | 0.0063 (0.59) | 0.22 (0.29) |
| RAM | 3.23 (0.55) | 7.97 (0.33) | 0.059 (0.82) | 2.39 (0.36) | 0.0063 (0.57) | 0.39 (0.28) |
| VAR | 3.87 (0.43) | 2.96 (0.79) | 0.035 (0.49) | 1.18 (0.42) | 0.0068 (0.61) | 0.21 (0.29) |
| VDL | 3.26 (0.43) | 7.75 (0.31) | 0.112 (0.68) | 2.50 (0.38) | 0.0055 (0.55) | 0.29 (0.26) |
| YZM | 1.92 (0.55) | 6.24 (0.25) | 0.564 (0.32) | 1.98 (0.41) | 0.0055 (0.57) | 0.29 (0.26) |

LEL, and NYP. Post-model-pre-data distributions for $N_t$ in the final time-step were updated on all estates. We use results from six estates (DLQ, CUL, HIR, NOG, NYP and VAR) as case studies and interpret these in more detail given the culling effort and number of foxes killed. We show DLQ (Fig 3) as an example of the results available for all estates (figures for the other 21 estates are shown in S5 Appendix).

Mean fox density on DLQ across the period data was recorded was 3.53 fox km$^{-2}$ (Fig 3B). The annual level of lamping effort on this estate was average (S1 Table) but was concentrated into the period just before and during the bird nesting period so was seasonally intensive. Across years, the mean weekly cull was 0.166 fox km$^{-2}$ wk$^{-1}$ and the high culling effort intensity on DLQ led to generally high suppression of fox density, especially during the nesting period when culling was targeted. Pre-breeding fox density was calculated as the mean of the posterior median fox density during February ($\bar{N}_t$). $\bar{N}_t$ was 52% of the posterior median carrying capacity, but $N_t$ was less than 50% of $K$ for considerable periods of time during the year. For short periods, e.g. following spring 1998 where nine cubs were removed at earths, $N_t$ was close to zero (Fig 3C). The posterior median for immigration rate onto DLQ was amongst the highest in the 22 estates, as were the values for carrying capacity and initial density, while rate of successful search was one of the lowest values estimated (Fig 3D, Table 3). The high estimates for both $N_t$ and $v$ were expected as it was known that there was no fox control at DLQ on adjacent land-holdings [65].

Mean fox density on CUL was 2.17 fox km$^{-2}$ (S5 Appendix: Figure A(b)). The level of culling effort was relatively low (S1 Table) but there was a constant removal of foxes during the year with a mean weekly cull of 0.062 fox km$^{-2}$ wk$^{-1}$. Pre-breeding $\bar{N}_t$ was 61% of $K$ (S5

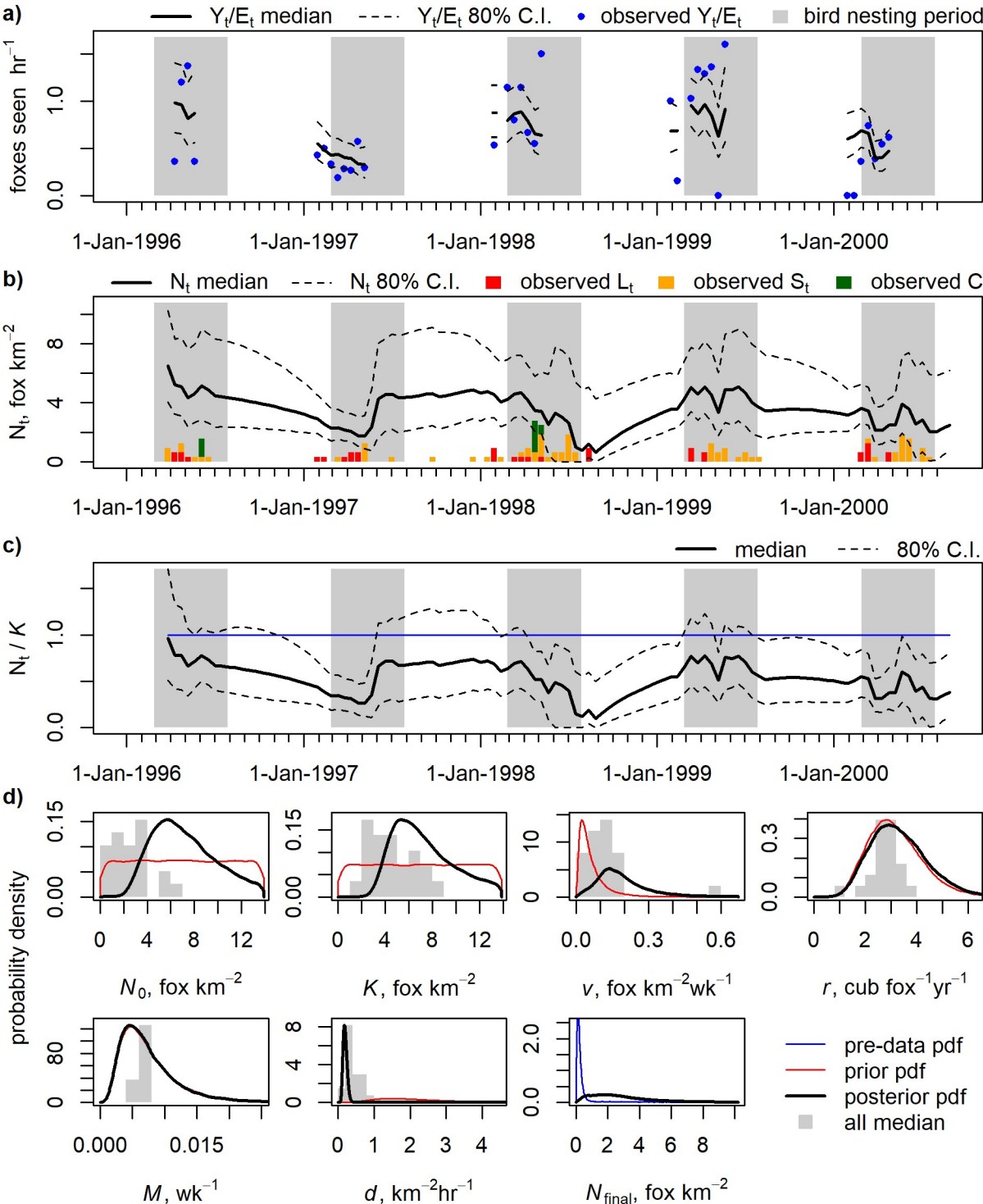

**Fig 3. Example results for DLQ.** Panel a) shows posterior fit of the model to detection rate ($Y_t/E_t$). Panel b) shows posterior estimates of bi-weekly fox density ($N_t$) in relation to the cull removed by different methods. Panel c) shows estimated $N_t$ as a proportion of carrying capacity (blue line denotes median population at posterior median $K$). Panel d) shows priors (or post-model-pre-data distribution) and marginal posteriors of $N_0$ (initial density), $K$ (carrying capacity), $v$ (immigration rate), $r$ (*per capita* birth rate), $M$ (instantaneous non-culling mortality rate), $d$ (rate of successful search), and fox density in the final time-step. Histograms in panel d) show posterior medians from all estates. In panels a-c) the bird nesting period (March-July) is shaded grey as a reference.

Appendix: Figure A(c)). The mean annual cull was greater than the mean fox density, but although fox density was suppressed relative to what would be expected in the absence of culling, this level of culling pressure was not enough to maintain a low fox population throughout the nesting period. The posterior medians for parameters on CUL were mid-range relative to the posterior medians estimated on the other estates (Table 3, S5 Appendix: Figure A(d)).

Culling effort on HIR was spread throughout the year but the annual lamping effort was the highest among all 22 estates (S1 Table). The mean weekly cull was high at 0.110 fox km$^{-2}$ wk$^{-1}$. The mean fox density on this estate was 0.92 fox km$^{-2}$ and the detection rates were low (S5 Appendix: Figure B(a,b)). Pre-breeding $\bar{N}_t$ was 39% of $K$, but suppression during the March-July gamebird nesting period was greater, as the mean of the posterior median fox density during this period was only 29% of $K$ (S5 Appendix: Figure B(c)). The posterior median for carrying capacity was low relative to other estates, but the values for the other parameters were in the mid-range (Table 3, S5 Appendix: Figure B(d)).

Mean fox density on NOG was 2.40 fox km$^{-2}$ (S5 Appendix: Figure C(b)). Culling effort was applied year-round and lamping effort was of average intensity (S1 Table). The mean weekly cull was 0.084 fox km$^{-2}$ wk$^{-1}$, though notably no cubs were killed at earths in any year. The fox population was well-suppressed during the winter and early part of the bird nesting period prior to cub recruitment, but because there was no control of cubs the population increased sharply at this time (S5 Appendix: Figure C(c)). Pre-breeding $\bar{N}_t$ was 44% of $K$. The posterior medians for parameters on NOG were mid-range relative to the posterior medians estimated on the other estates (Table 3, S5 Appendix: Figure C(d)).

Mean fox density on NYP was 3.73 fox km$^{-2}$ and detection rates were up to twice those seen on other estates (S5 Appendix: Figure D(a,b)). Culling effort was applied throughout most of the year but at a relatively low level (S1 Table). The mean weekly cull was 0.08 fox km$^{-2}$ wk$^{-1}$. Suppression of the fox population was poor, and for winter 1997–98 $N_t$ spent five months above $K$ (S5 Appendix: Figure D(c)). Pre-breeding $\bar{N}_t$ was 90% of $K$. The posterior median for immigration rate onto NYP was high relative to the posterior medians estimated on the other estates (Table 3, S5 Appendix: Figure D(d)).

Detection rates on VAR were low and mean fox density was 1.04 fox km$^{-2}$ (S5 Appendix: Figure E(a,b)). Culling effort was maintained year-round. Lamping effort averaged almost 540 hours per year, the highest value for any estate, although this was not a particularly high lamping effort relative to estate area (S1 Table). The mean weekly cull of 0.035 fox km$^{-2}$ wk$^{-1}$ was low compared to HIR where fox density was similar, but the suppression of $N_t$ relative to $K$ was very high, especially from 1997 onwards (S5 Appendix: Figure E(c)). Pre-breeding $\bar{N}_t$ was 40% of $K$, but suppression was greater during the nesting period at 27% of $K$. The posterior median for immigration rate was low relative to other estates, and *per capita* birth rate was the lowest estimate (Table 3, S5 Appendix: Figure E(d)).

## Population suppression

Across all estates, estimates of pre-breeding fox density ranged from 0.92 to 4.43 fox km$^{-2}$ (Fig 4). Pre-breeding fox density as a proportion of posterior median carrying capacity ($\bar{N}_t/K$), ranged from 20% to 90% of $K$. This indicated that, to varying degrees, culling had suppressed the fox population on all estates at this time of year. Fox density during the critical March-July gamebird nesting period (week 10-week 30) was calculated as the mean of the posterior median fox density during this time. Nesting period $\bar{N}_t$ ranged from 0.73 to 4.69 fox km$^{-2}$ (Fig 4), with $\bar{N}_t/K$ ranging from 27% to 78% during the nesting period. Relative to pre-breeding density, only 7/22 estates increased suppression of density during the nesting period.

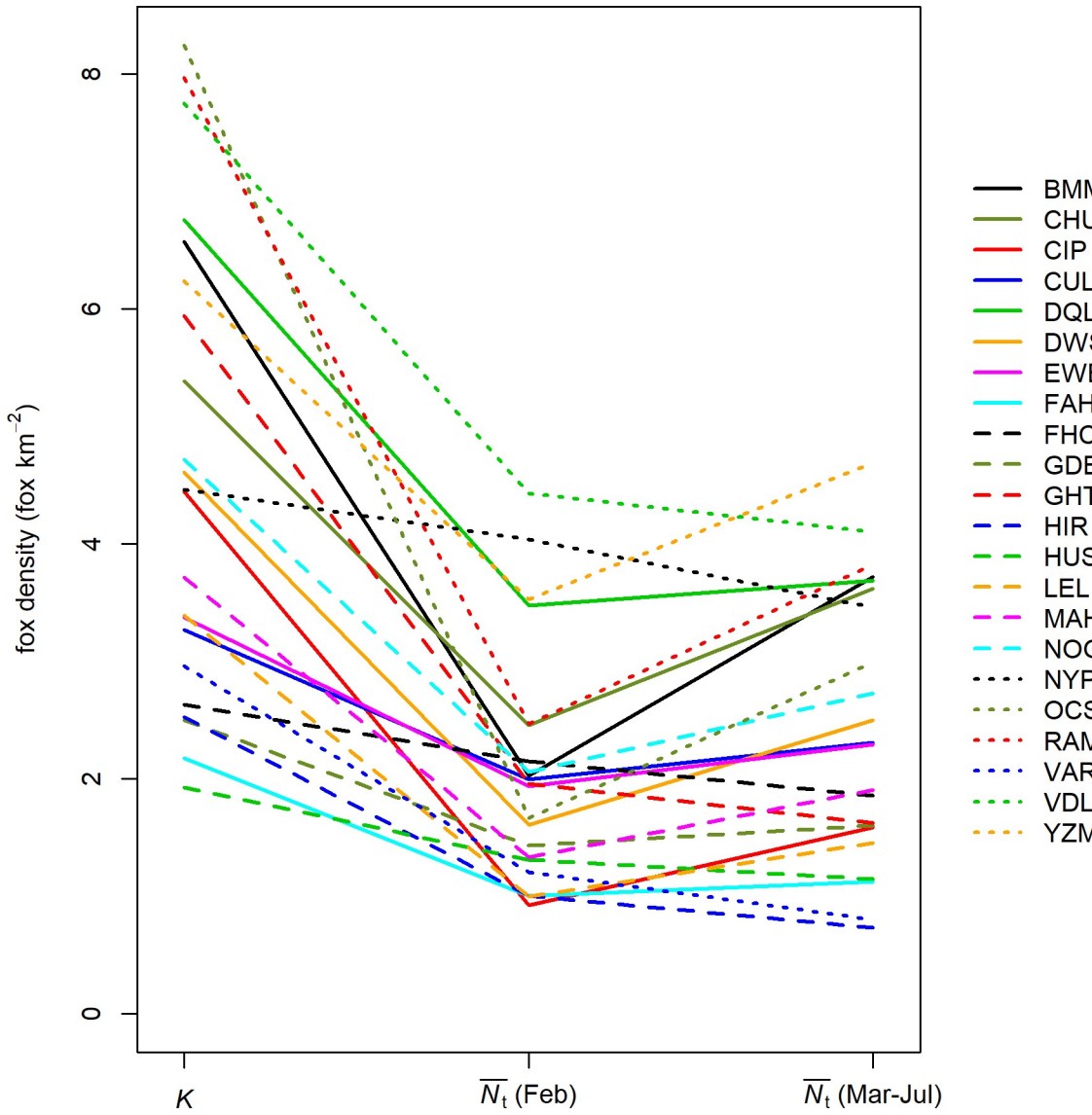

**Fig 4. The knock-down effect of culling determined by the estimated fox density at significant times of year, relative to estimated carrying capacity.** The across-year mean of posterior median fox density ($\bar{N}_t$) on each estate during the pre-breeding period (February) and the gamebird nesting period (March-July) is compared to the posterior median of carrying capacity ($K$) specific to each estate.

## Mortality comparison

During the period covered by the data, there were 2,702 foxes culled across the 22 modelled estates (1,469 by lamping, 958 by other methods, and 275 cubs at earths). The contribution of estimated non-culling mortality to total mortality was small relative to the culling mortality on all estates (Fig 5). On most estates the cull exceeded estimated initial density within 6–12 months, and on more than half of the estates annual culling mortality exceeded the estimated carrying capacity. Annual non-culling mortality was on average around 16% of culling mortality. Cumulative non-culling mortality typically exceeded carrying capacity after about four years, which given our model assumptions implies a population turnover of four years in the absence of culling.

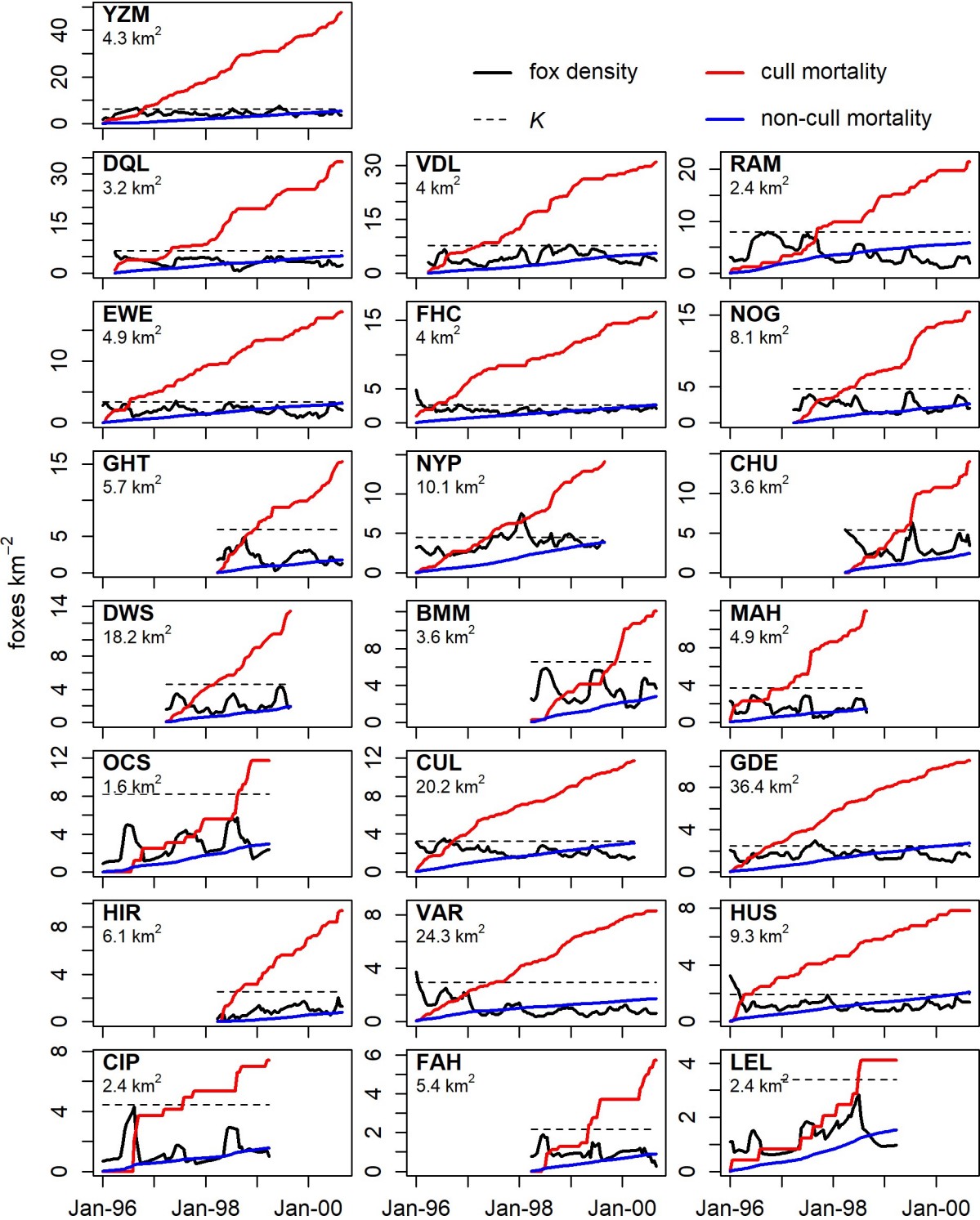

**Fig 5. Comparison of the cumulative culling mortality and estimated cumulative non-culling mortality implied by the posterior median of _M_ on each estate over the range of the contributed data.** Posterior median fox density and carrying capacity of each estate are shown. The scale of mortality (shown in dead foxes km$^{-2}$) differed so estates are ordered by row from top-left to bottom-right by decreasing total mortality. On most estates the cull dwarfed estimated non-culling mortality and exceeded estimated initial density within 6–12 months.

## Sensitivity analyses

Parameter estimates were robust to assumptions related to timing of weaned cub recruitment (Fig 6) and there was no effect on $N_t$ (Fig 7A). We found only a minor effect on some estates of assuming that immigration is not a year-round process. Fox density estimates under a seasonal immigration assumption were unaffected on CUL, NOG or DLQ (Fig 7A). Compared to the constant immigration reference case, differences in $N_t$ estimates on NYP and VAR (lower) and HIR (higher) rarely exceeded one fox per km², caused by differences in estimates of $v$, $K$ and $d$ on these three estates (Fig 6). Conclusions about population suppression on NYP, VAR and HIR were unaffected due to the different estimates of $K$. The effects of alternative assumptions about priors were restricted to the marginal posterior of the parameter being assessed (Fig 6) and did not affect estimates of $N_t$ (Fig 7B). The exception was the use of a lower upper bound on the uniform prior for $N_0$, which resulted in lower $N_t$ estimates on DLQ (Fig 7B) because the $N_0$ posterior was truncated at this value (Fig 6). The truncated posterior indicated the alternative prior was an unsuitable choice.

The mean estimates of empirical process error standard deviation were all lower than 0.2 (assuming Poisson observation errors), with the mean across estates equal to 0.12 (S1 Fig). This indicated that the realised process errors were smaller on average than assumed in the model. Sensitivity to process error assumptions are presented for DLQ and VAR only, as these represented the range of available detection rate time series on those estates which met the minimum data requirements: seasonal (DLQ) to near-continuous (VAR). The use of $\sigma_p$ values smaller than the 0.2 reference case led to convergence problems when modelling DLQ data, and to unfeasibly high density estimates on both estates. Due to lower estimates of the rate of successful search, use of fixed values for $\sigma_p$ of 0.05 and 0.1 resulted in the mean of posterior median fox density along the length of each time series to be 14% and 41% higher on DLQ (S2 Fig); and 106% and 51% higher on VAR (S3 Fig).

Use of an informative prior for $\sigma_p$ on DLQ resulted in mean $N_t$ estimates 47% higher than the reference case (S2 Fig), with the posterior median for $\sigma_p$ close to the prior median of 0.05. This highlighted convergence problems when $\sigma_p$ was small, as using an informative prior centred around 0.05 produced markedly different results to those from fixing $\sigma_p$ at the same value. Convergence problems were reduced by using a vague prior on $\sigma_p$, as the posterior median was increased to 0.30 and resulted in mean $N_t$ estimates 18% lower than the reference case (S2 Fig). On VAR, there was minimal effect of using an informative prior for $\sigma_p$ as the posterior median of 0.16 was close to the reference case where $\sigma_p$ equalled 0.2 (S3 Fig). However, use of a vague prior for $\sigma_p$ resulted in a posterior median of 0.52, a considerably higher value than assumed in the reference case, and the highest among all estates. Mean $N_t$ estimates were up to 32% lower, caused by different estimates of $K$, $v$, $r$ and $d$ (S3 Fig). However, $N_0$ was estimated to be almost four times $K$, which on an estate where there was historic culling seemed highly unlikely. Overall, the problems encountered using alternative process error specifications supported our decision to fix $\sigma_p$ at 0.2.

## Discussion

Predator control is often used to improve the conservation status of threatened prey species or to increase the abundance of game species for hunting [11]. However, without quantitative data, we cannot assume that local scale culling on restricted-areas will reduce predator density, let alone achieve its aims. Indeed, the responses of the population to removals, e.g. through compensatory immigration, may result in control being ineffective and inefficient. Wildlife managers employing lethal control of foxes on restricted areas therefore need methods to assess the impact of culling on fox density within the managed area. In addition, as predation

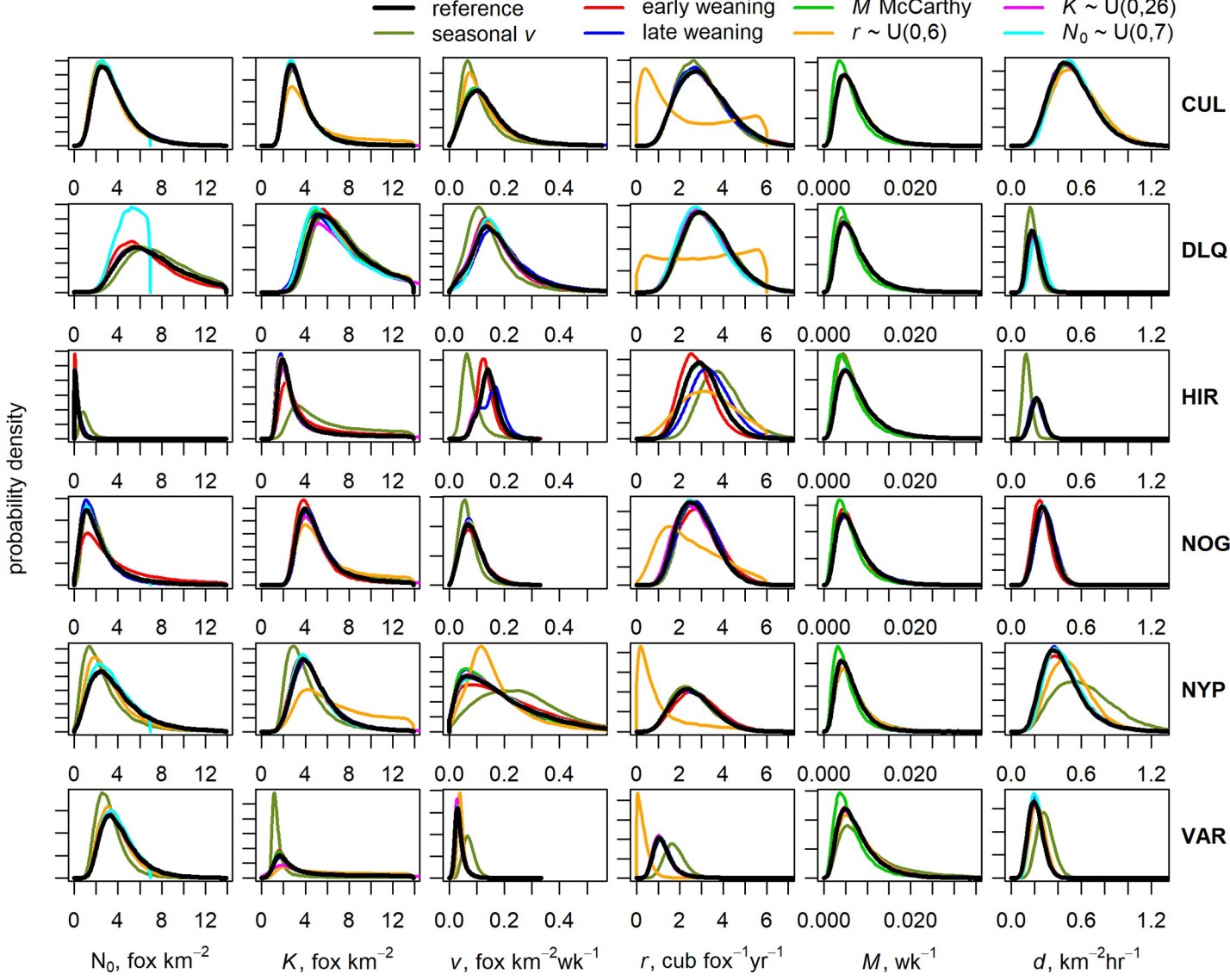

**Fig 6. Sensitivity of marginal posterior distributions for parameters (columns) from six estates (rows) to different structural assumptions and prior distribution specifications.** The reference case distributions shown are from the informative prior model. These are compared to the distributions obtained under assumptions about the seasonality of immigration and timing of cub recruitment, and specification of prior distributions for $M$, $r$, $K$ and $N_0$. The y-axes are not labelled for clarity.

risk of ground-nesting birds reflects the variation in density of foxes in the local area, an evaluation of the effectiveness of culling must involve quantifying fox density [66].

We describe a model for estimating the demographic parameters and density of a fox population in the context of restricted-area culling, using data on detections and culling effort collected during the culling process. The generalised depletion model developed incorporates non-culling mortality, density-dependent cub recruitment and immigration. These processes were effectively represented and the latent states were reliably estimated using a Bayesian state-space modelling framework and informative priors for key parameters; this finding is consistent with experience from other state-space population dynamics models [67]. State-space models have been used to understand the dynamics of exploited terrestrial populations,

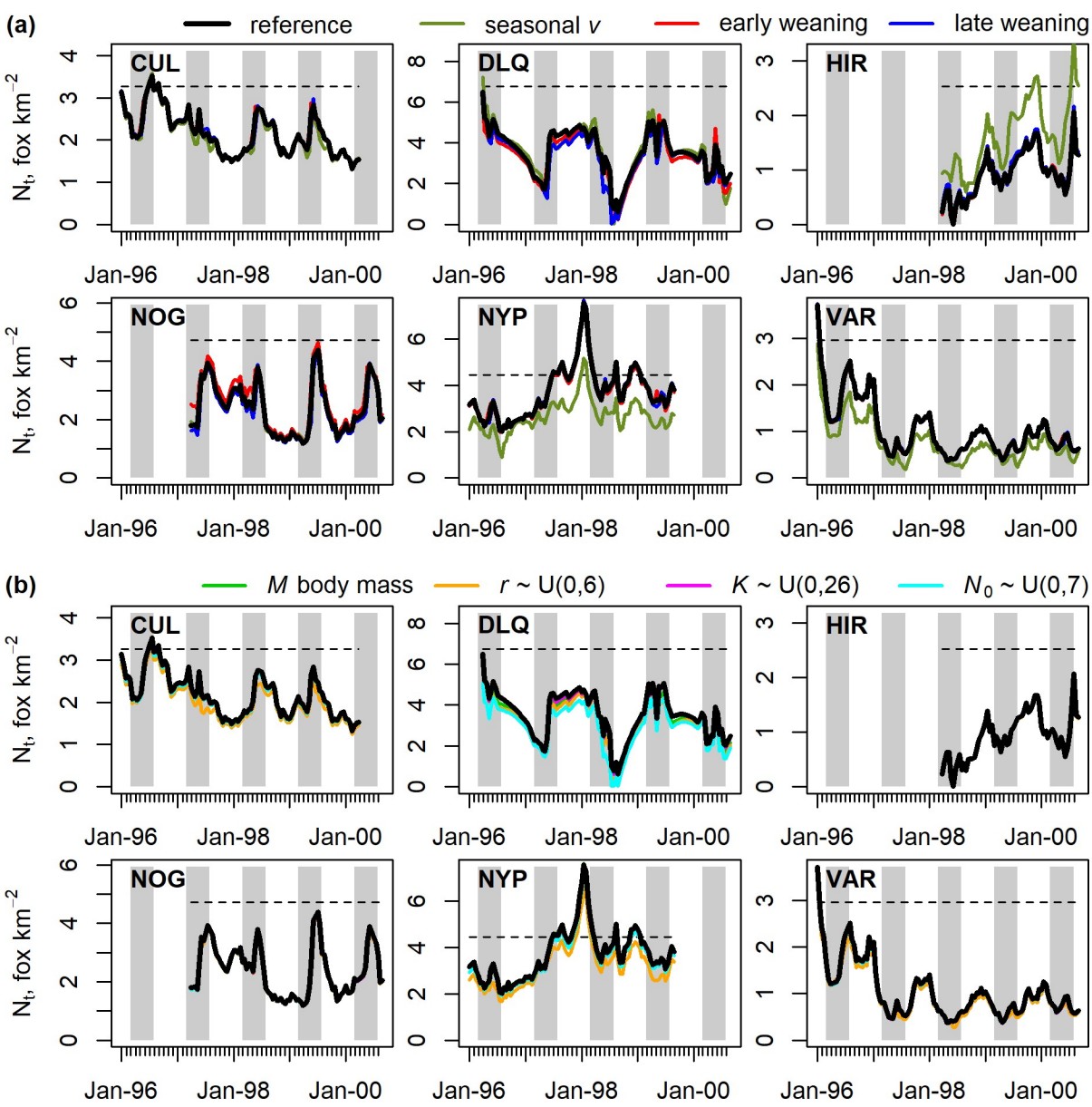

**Fig 7. Sensitivity of posterior median fox density on six estates to different structural assumptions and prior distribution specifications.** Panel (a) shows sensitivity to structural assumptions about the seasonality of immigration and timing of cub recruitment. Panel (b) shows sensitivity to specification of prior distributions for $M$, $r$, $K$ and $N_0$. Line colours for each assumption are shown above each set of charts. Reference case (black line) is from the informative prior model. The dashed lines show the posterior median carrying capacity for the reference case.

e.g. black bear (*Ursus americanus* [68]) and greater snow goose (*Chen caerulescens atlantica* [69]). State-space models have also been used to model fox population dynamics [70], but not in the context of understanding the response to culling. We applied our model to data from 22 estates, which although not representative of circumstances across Britain, being in only a few regions and landscapes, resulted in parameter and density estimates that were nevertheless revealing about the relationships between the cull and within-year fox population dynamics.

## Parameter estimates

Posterior distributions for model parameters showed considerable updates from the priors on most estates. Estimates of immigration rates, *v*, were mostly updated to median values higher than the prior median for *v*. This can be explained by the location of the modelled estates, which were predominantly in arable and pastural landscapes, and not uplands. Immigration rate estimates in arable and pastural landscapes [17] were higher than the median of the posterior predictive distribution upon which the prior was specified. This was not unexpected, as the prior accounted for all landscape types, including upland landscapes where fox density is known to be low, but nevertheless the estimated immigration rate on some estates was particularly high. The immigration rate parameter is a maximal rate, reflecting the maximum number of foxes that would move onto the estate if the fox density was maintained near zero throughout the year. The highest estimate for *v*, on YZM, was equivalent to over 29 foxes km$^{-2}$ yr$^{-1}$. This meant that despite the high intensity of culling effort and the large number of foxes culled on YZM, the suppression of fox density relative to carrying capacity there was only modest. This estate was in a pastural landscape, but the location can explain the high *v* estimate as it had a large urban area on one boundary and a forested national park where there was no intensive fox control on the other. Both areas could act as source populations of foxes.

The degree of posterior updating for *per capita* birth rate, *r*, was variable but on most estates the median estimates were similar to or lower than the median of the informative prior, indicating that the prior was having an influence on the results. Indeed, posteriors for *r* were sensitive to the choice of prior, but fox density estimates were similar using either prior because the central tendency under both priors was similar, at around 3 cubs fox$^{-1}$yr$^{-1}$. The posteriors for *r* on HIR showed the most similar results under either prior choice. This is probably because the large number of cubs killed at earths on this estate provided more information about how many cubs must have been born. A consequence of using litter size per female to specify the prior for *r* (S3 Appendix) was that posterior estimates of *r* could be updated to lower values relative to the prior either because the *per capita* birth rate was lower on an estate or because there was within-earth non-culling mortality of cubs. Without information on within-earth mortality it was not possible to separate these two effects. This is not a problem *per se*, as the modelled process was the recruitment of weaned cubs, but interpretation of the parameter estimates as *per capita* birth rates for comparison with other studies must be performed with care.

The lowest estimate for *r* was 1.16 cub fox$^{-1}$yr$^{-1}$, on VAR. Because this is so much lower than expected birth rate, it suggests significant within-earth non-culling mortality before cubs were weaned on this estate. There were very few cubs culled at earths on this estate despite a high level of control by lamping and other methods. The recorded cull of cubs may have been small either because little effort was put into it or because there were few cubs to cull; this is unknown. The use of fumigants to kill cubs within the earth cannot be ruled out, although no product was authorised for this purpose during the period in question. The fox population on this estate was low and appeared to be heavily suppressed. It could be that due to the intensity of control vixens were being killed following birth of any cubs, causing mortality of the dependent cubs within the earths. Culling during the spring and summer, while especially beneficial for prey species due to the reduction in cub food requirements, does carry a welfare cost through failure to locate and destroy orphaned cubs [18]. VAR was a very large estate (24.3 km$^2$), so considerable effort would have been required to find and monitor all the potential breeding earths across it.

Carrying capacity, *K*, was updated from the vague prior on all estates, and the posterior median varied fourfold. The range of these estimates of *K* in rural fox populations in Britain closely mirrored the range estimated across different sites in rural France (0.9–8.3 fox km$^{-2}$

[7]). The CV of *K* estimates varied greatly, suggesting that data from some estates were more informative than others. The simulation-estimation analysis found that carrying capacity was better estimated in datasets where fox density spent more time close to *K*. Some of the highest CV values were from those estates where suppression of density relative to *K* was high, i.e. RIH and VAR, where density was maintained well below any plausible value of carrying capacity and so estimates were expected to be more uncertain. In the absence of a covariate for *K*, e.g. habitat or resource availability, use of an informative prior could reduce this uncertainty and reduce posterior correlations with *K* in future applications. The marginal posteriors of *K* from this study could be synthesised across all estates for use as an informative prior, given appropriate consideration of the limited spatial representation of the estates modelled.

Posteriors for the rate of successful search, *d*, were consistently updated to lower values than the prior. The most reasonable explanation for lower estimates of *d* seems to be that lamping behaviour differed from that assumed under the prior. The informative prior was specified using data from a distance sampling survey conducted from standing in the back of a vehicle travelling on surfaced roads [44]. Within shooting estates, travel will mostly be on unsurfaced tracks, and progress will be slower. Lamping may also be performed cross-country from a quad bike or on foot, when the speed of travel is likely to be considerably slower; also, the field of view will be smaller because the observer's viewing position is lower. Since these data were recorded, thermal-imaging and night-vision have become more popular amongst gamekeepers compared to traditional spotlights, likely increasing the probability of detecting foxes. As the rate of successful search is the product of detection probability, field of view, and speed of travel, these new technologies are expected to lead to increased rates of successful search.

Non-culling mortality rate showed the least updating, supporting the findings from the simulation-estimation analysis that *M* is weakly identifiable using this model. Depletion model applications in the fisheries literature often choose to fix *M* at some value to remove such identifiability problems [35,71]. Given the lack of posterior updating of *M*, with the posteriors closely reflecting the prior, there is an argument that we should fix *M* here. However, fixing parameter values does not admit any parameter uncertainty, so use of an informative prior to both reduce the identifiability problems and incorporate parameter uncertainty is preferable [72]. While the posteriors for *M* were sensitive to the choice of informative prior, this did not greatly affect fox density estimates. There was no relationship between posterior median estimates of *M* and estate size, indicating the implicit emigration rate component of *M* was minimal as it was expected that smaller estates would have higher values of *M* if emigration was large.

### Effect of culling on fox density

Previous studies have concluded that restricted-area culling is not greatly effective at reducing local fox density [6,7,66]. The estimates of fox density here show that culling suppressed the population relative to estimated carrying capacity on all estates, with pre-breeding density on average 47% of *K*. Because growing cubs require more food than adults, this degree of suppression would have more than halved the food requirements of the fox population relative to the no-cull situation. In respect of reducing predation risk, the culling approach exemplified by these estates must be considered a success, given the experimental evidence that effective predator control translates into improved breeding success and population trend of ground-nesting birds [3,4]. All estates had late-winter (February) density below carrying capacity, and in most cases well below. Despite some large culling efforts and annual bags greater than fox density, few estates maintained fox density at a low level throughout the year. Under the most likely

reconstruction of fox density, as determined by the posterior median, no estate had zero fox density throughout the bird nesting period between March and July, although credible intervals were bounded at zero on four of the 22 estates. On only one estate (HIR) did minimum fox density translate to an abundance of less than one fox within the estate, and only for a single time-step. On every other estate there was always at least one fox present.

Estates with enough data to fit our model were those with intensive culling efforts and were thus likely to play the role of sinks in regional population dynamics, and this is supported by the short time-scale in which the accumulating cull could exceed estimated carrying capacity (Fig 5). High immigration rates relative to removal rates appear to have made it difficult to further reduce within-estate fox density, as has been suggested previously [5,7,31]. In our model, recruitment of immigrants into the within-estate population is density-dependent, i.e. lowering within-estate density frees up more capacity for recruitment. However, the underlying immigration rate and thus the rate of replacement of each culled fox is modelled as a constant $v$ because it is assumed to be contextual. The rate of removal per unit effort of each fox present is also realistically modelled as a constant. This will be experienced by the gamekeeper as a falling catch-per-unit-effort as density falls, simply because there are fewer foxes present. Further suppression of fox density would require more intensive culling effort to increase culling mortality rate and thus overcome the rapid replacement rate. This may not be practicable. The effect of immigration rate on the level of control required can be seen by comparing the results on HIR and VAR. The mean fox density on these estates was similar at around 1 fox km$^{-2}$ and so was estimated carrying capacity at around 3 fox km$^{-2}$. To maintain the same fox density, the annual cull on HIR had to be twice as large as on VAR due to higher immigration and *per capita* birth rates. With a comparatively low immigration rate onto the estate, VAR achieved the most consistently low fox density. Because of the large size of the estate this came at a cost of some 540 hours of lamping per year, which was by far the biggest investment per estate although by no means the most intensive in terms of hours per km$^2$ per week.

The number of foxes killed was shown to be a poor indicator of culling effectiveness. For instance, VAR had one of the lowest culls of foxes per km$^2$, but the year-round effort appeared highly effective at suppressing density. The low number killed simply reflected a lower density of foxes available to cull. This ambiguity is a problem with the use of bag data where effort is not measured [18,73,74]. The scale and frequency of effort were better predictors of effectiveness. DLQ, GHT, HIR and VAR all used above-average levels of lamping effort. These estates achieved some of the lowest fox densities and greatest levels of population suppression. However, considering effort alone is not enough as FHC and YZM also used high levels of effort but failed to achieve much impact on fox density because of high immigration rates. Thus, the model-based approach is necessary to understand how culling effort and population processes, e.g. immigration, interact to determine the impact of culling on population density within each estate.

The relatively moderate suppression of fox density achieved through the pre-breeding and bird nesting periods on some estates does not necessarily mean that control was ineffective on those estates. We have considered impact in terms of predation pressure on wild ground-nesting birds, although the actual aims of each estate may have differed from this. In the wild bird context, DLQ is of special interest because it was the site of the Allerton Project, a high-profile demonstration of 'wildlife-friendly farming' using game management techniques [65,75,76]. Detailed monitoring of small game populations at this site, and on comparison sites nearby with no predator control, contributed to evidence that common predators can limit the density of some prey species [76–78]. The contribution of foxes to predation will therefore be of interest to many as a case study. Beginning in 1993, control of foxes and corvids was implemented during the spring and summer. Autumn counts of pheasants and hares showed that these prey

populations had grown significantly by the start of 1996 and remained high through 2000 until predator control was stopped on the estate in 2001. Annual variation in pheasant and hare breeding success was previously ascribed to weather effects [76], but was also consistent with the apparent success of fox culling in each year. Although pre-breeding fox density was suppressed to 52% of carrying capacity, and fox density through the nesting period was never zero, flourishing prey populations offer support for the notion that the strategy of intensive fox culling only during the spring and summer was achieving its aims.

Heydon et al. [14] have previously shown that because of regional patterns of fox culling intensity, landscape alone is not a satisfactory predictor of regional fox density in Britain. Only six estates in this study (CIP, FAH, HIR, LEL, RAM, VAR) had estimates of pre-breeding density within a contemporary range of estimates derived from faecal density counts stratified on the basis of landscape [79]. All six of these estates achieved above-average suppression of the fox population relative to estimated carrying capacity. For the other 16 estates, estimates of pre-breeding fox density were generally higher than predicted based on landscape type, despite being suppressed relative to carrying capacity. This may reflect the greater densities of game and other prey on estates managed for shooting, due to habitat management and control of predators, and supplemented by release of captive-bred gamebirds. On estates where releasing forms the basis of the shoot, densities of gamebirds may be considerably higher and the birds more naïve toward predators than wild stocks. Game management may thus create a 'honeypot' of higher prey availability, raising within-estate carrying capacity and possibly immigration rate. Understanding these relationships is the focus of subsequent work.

### Influence of model assumptions

We assumed that immigration was at a constant rate year-round, so that replacement could occur whenever a fox was culled. Fox migration is generally considered a seasonal process with a peak dispersal period in October-January [36]. However, in a population subjected to culling, perturbations to territorial structure can lead to rapid replacement at any time of year, either by itinerant foxes filling the space created or by neighbouring territory-holding foxes absorbing the undefended area [18,31]. Intensive telemetry data shows that foxes will exploit newly undefended areas adjacent to their own territory after only one week (unpublished data, GWCT). A vixen will also move her cubs to a different location that may be up to 1.5 km away [31,37]. As gamekeepers can remove cubs only from earths within their estate, all these movements will be perceived by the gamekeeper (and in our models) as immigration of foxes across the estate boundary and onto the culled area. Across all estates in the present study, more foxes were culled during August-October than at other times; however, this does not necessarily reflect seasonal dispersal because at this time of year lamping is easier following the harvest of arable and fodder crops, so lamping effort was at its highest, and any cubs born locally would have recruited into the observable population.

Alternative assumptions about the seasonality of immigration rates had minimal effects on estimated fox density. If immigration occurred only during the dispersal period, the expected effect of assuming constant immigration would be that the immigration rate of foxes would be underestimated during the dispersal period. In the case of all other parameters being constant, this would lead to underestimation of fox density and overestimation of the effectiveness of culling at this time of year, and *vice versa*. Although differences in fox density were found on some estates under a seasonal immigration assumption, resulting from differences in the $v$, $K$ and $d$ parameter estimates, the direction of these differences was not consistent between estates. Of the six estates examined, the seasonal estimates of $v$ were higher relative to the

constant model on NYP and VAR, but lower on all other estates. The assumption that immigration rate is constant over time also implies that carrying capacity within each estate, e.g. due to food availability, and availability of foxes from neighbouring source areas were unchanged throughout the five-year period. These details and their possible effect on conclusions are unknown.

The distribution of weaned cub recruitment dates used in the model adequately captured the full variation found in reality, as there was minimal effect of the mean recruitment date being two weeks earlier or later. The onset of breeding occurs in late-winter and is correlated with day length, starting earlier at more southerly latitudes [80]. Across the range of latitudes in Britain (50˚-59˚ N), ovulation date varied by around three weeks [80,81]. The two fox populations used to establish the distribution were in Wales and south-east England [37]. These two regions are on similar latitudes, so given that most of the estates were within this range (51˚-53˚ N covered all but one estate) we expected minimal differences in timing of breeding events.

In state-space models, process errors account for unexplained variation in animal density due to environmental and demographic stochasticity. To reduce extrinsic identifiability problems due to confounding between the strength of density-dependence and process error [28], we fixed the standard deviation in process errors at 0.2. Given values for $\sigma_p$ smaller than this, the fit of the model failed to capture the variation in detection rate, with the credible interval for predicted detection rate omitting many of the observed data points. Consequently, it was more difficult to achieve model convergence under smaller values of $\sigma_p$ and reconstructed fox density took unrealistically high values. The empirical estimates of $\sigma_p$ suggest that while the model required a value of 0.2 to converge onto the joint posterior, the realised variation in $N_t$ caused by process error was within the range permitted by the model. We examined use of an informative prior for $\sigma_p$ but this also led to problems with model convergence, so we did not explore alternative weakly informative priors, e.g. half-$t$ [70]. When $\sigma_p$ was estimated using a vague prior there were some very high estimates on some estates. Elevated estimates of process and observation variance may suggest errors in model specification that could lead to biased estimates [42]. However, the highest estimate of $\sigma_p$ was on VAR, an estate on which lamping was conducted almost every week for the entire length of the five-year period; this result is consistent with the finding of a study on ungulates where Bayesian process error variance increased with time series length [82].

Density-dependence is a key consideration in the management of exploited species [83]. We incorporated density-dependence into our model by assuming there is a carrying capacity density at which resource availability limits the fox population, which is reasonable as higher fox densities are found where more food is available, e.g., in urban environments. There is also abundant evidence that fox productivity is suppressed when density is high relative to food resources [13,18]. The assumption in using logistic terms to model this density-dependence is that there is a linear decline with density in the rates of population growth by cub recruitment or immigration. For territorial animals such as foxes, a non-linear relationship may be more realistic as growth is likely to be relatively unaffected by increasing fox density while there is vacant territory space, but once crowding occurs at higher densities the growth will slow rapidly as the population nears carrying capacity. The theta-logistic model allows for this convex relationship through an extra shape parameter which determines the functional form of density-dependence [84,85]. However, the shape parameter is difficult to estimate and can lead to unreliable inferences [86]. We could have expanded the set of model structures explored to further examine the robustness of our fox density reconstructions, but as our model already appears to be on the limits of identifiability, we did not consider options with additional parameters, such as the theta-logistic model.

If appropriate informative priors had not been available to reduce the identifiability problems, particularly relating to estimation of $M$, one option we considered was a simplification of the depletion model structure which combined non-culling mortality and immigration rates into one parameter, i.e. apparent survival. This option was not considered further because 1) immigration is clearly a key process in the dynamics of a culled population, requiring explicit estimation, 2) it would then not be possible to assess the relative importance of non-culling mortality to total mortality, and 3) modelling density-dependence becomes very complicated. An approach which would provide identifiable estimates of all the demographic parameters without need for informative priors is integrated population modelling [87,88], but in addition to count data such as those in our fox culling data, mark-recapture data are required. This would require a suspension of culling to allow foxes to be captured for tagging and release, which seems unlikely to be tolerated; or the use of genetic identification methods, with associated sampling and cost issues.

We assume the rate of successful search is constant over time, but non-constant detectability can be a source of error when using relative abundance indices such as detection rate [20,34]. Foxes may exhibit changes in behaviour, e.g. due to weather, breeding season, or in response to previous control. There may be seasonal variation in detectability, e.g., due to crop cover changing during the summer. This could potentially lead to biased estimates of fox density during the summer, but without covariate data on the habitat types in which foxes were seen, or a field experiment to determine how detection probability varies seasonally, it was necessary to assume constant detectability. To avoid any bias, detection rate data from the summer period could have been omitted, but as data during this period is necessary for reliable estimation of cub recruitment, this was not a real option. In any case, the amount of lamping effort was lowest during this period (S4 Appendix: Fig A)–presumably as vegetative cover reduced visibility–so we can reasonably assume that gamekeepers went lamping only if there was a fair chance of detecting foxes.

In these data, the probability of a fox being killed on detection was only 30%, raising speculation that foxes which had previously been shot at might exhibit spotlight avoidance behaviour, which would cause detectability to decrease over time. However, many foxes will be detected in locations where a safe shot is not possible, and this too will result in a low probability of killing on detection. Hence the proportion of foxes shot at and missed may be quite small. We do not know the number of shots fired per detection. Heydon et al. [14] could find no evidence for 'lamp-shyness' in regions with intensive culling by spotlight and rifle.

By assuming that handling time was zero, the model also assumed that $d$ does not vary with fox density. Failure of this assumption would cause the model to make incorrect predictions. Handling time represents the effective search time lost per fox encountered, e.g. due to time spent determining if a shot is safe, collection of shot foxes for age or sex determination, or refractory time caused by scaring of foxes within hearing distance of each shot taken, and therefore sets an upper limit on the number of possible fox detections for a given lamping effort. However, unless assumptions can be made about the handling time and its variability among gamekeepers, it is an additional parameter to estimate. In any case, for fox populations at those densities found in Britain, the effect of handling time is minimal [44].

## Conclusions

The effectiveness of restricted-area culling on local fox populations cannot be measured simply by the number of foxes killed, and the culling effort that reduces fox density in one situation may be insufficient elsewhere because of local circumstances. Our modelling shows that, to

varying degrees, culling on all 22 shooting estates suppressed fox numbers relative to estimated carrying capacity within the estate. Rapid replacement of culled foxes by immigration from neighbouring sources meant that continuous effort was required to maintain this effect year-round. Fox density within each estate typically peaked in late summer. During the critical March-July period when avian prey species are nesting, and fox food requirements are also elevated by breeding, lamping effort on many estates was reduced and the successful use of alternative culling methods was highly variable among estates. Consequently, a minority of estates maintained the low fox density achieved in February throughout the March-July period, although mean density through this period was below carrying capacity in all cases, and in most cases well below. We conclude that when suitably timed and sufficiently intensive, culling can effectively reduce fox density–and therefore food requirements–through an ecologically critical period. Our modelling approach offers a means of adaptive management tailored to local circumstances, to optimise the balance of conservation benefits against economic and animal welfare costs.

## Supporting information

**S1 Appendix. Timing of fox breeding events.**
(PDF)

**S2 Appendix. Choice of process error standard deviation.**
(PDF)

**S3 Appendix. Development of an informative prior for *per capita* birth rate.**
(PDF)

**S4 Appendix. Simulation-estimation analysis.**
(PDF)

**S5 Appendix. Fox density and parameter estimates from modelled estates.**
(PDF)

**S1 Fig. Histogram of the mean estimates of the empirical process error standard deviation from the 22 estates.** These realised values were obtained by calculating the standard deviation in process errors across the time series for each MCMC chain iteration and summarising the mean of those. The fixed value used in the model was 0.2.
(TIF)

**S2 Fig. Results for DLQ showing sensitivity of a) posterior median fox density and b) marginal posterior parameter estimates to specification of the process error standard deviation, $\sigma_p$.** The values of $\sigma_p$ were either fixed at 0.05, 0.1, or 0.2, or were estimated using either a lognormal prior distribution with median of ln(0.05) and CV of 0.2 or a uniform prior with lower and upper bounds of 0.001 and 1.0, respectively. The reference case, where $\sigma_p$ is fixed at 0.2, is shown in bold.
(TIF)

**S3 Fig. Results for VAR showing sensitivity of a) posterior median fox density and b) marginal posterior parameter estimates to specification of the process error standard deviation, $\sigma_p$.** The values of $\sigma_p$ were either fixed at 0.05, 0.1, or 0.2, or were estimated using either a lognormal prior distribution with median of 0.05 and CV of 0.2 or a uniform prior with lower and upper bounds of 0.001 and 1.0, respectively. The reference case, where $\sigma_p$ is fixed at 0.2, is shown in bold.
(TIF)

**S1 Table. Estate area, number of contributed weeks' data, number of weeks of lamping effort and lamping effort per km$^2$ in each year.**
(PDF)

**S1 File. BUGS model code.**
(TXT)

**S1 Dataset. Fox culling records used to model restricted-area population dynamics.**
(CSV)

## Acknowledgments

We are indebted to the 74 gamekeepers who for several years patiently recorded the data drawn on for this analysis. We thank Carl Walters for valuable comments on the chapters of TAP's thesis that formed the basis of this manuscript. We also thank Michael Scroggie and an anonymous reviewer for providing insightful comments that greatly improved this manuscript.

## Author Contributions

**Conceptualization:** Tom A. Porteus, Jonathan C. Reynolds, Murdoch K. McAllister.

**Data curation:** Tom A. Porteus.

**Formal analysis:** Tom A. Porteus.

**Funding acquisition:** Jonathan C. Reynolds.

**Investigation:** Tom A. Porteus.

**Methodology:** Tom A. Porteus, Jonathan C. Reynolds, Murdoch K. McAllister.

**Resources:** Jonathan C. Reynolds.

**Software:** Tom A. Porteus.

**Supervision:** Jonathan C. Reynolds, Murdoch K. McAllister.

**Validation:** Tom A. Porteus.

**Visualization:** Tom A. Porteus.

**Writing – original draft:** Tom A. Porteus.

**Writing – review & editing:** Tom A. Porteus, Jonathan C. Reynolds, Murdoch K. McAllister.

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
