## [Decision Letter · Decision Letter 0]

4 Sep 2019

PONE-D-19-17263

Population dynamics of foxes during restricted-area culling in Britain: advancing understanding through state-space modelling of culling records

PLOS ONE

Dear Dr. Porteus,

Thank you for submitting your manuscript to PLOS ONE. After careful consideration, we feel that it has merit but does not fully meet PLOS ONE’s publication criteria as it currently stands. Therefore, we invite you to submit a revised version of the manuscript that addresses the points raised during the review process.

Both reviewers highlighted the quality of the paper and the use of a nice approach to investigate the impact of culling on the population dynamic of a predator. They only have minor request to revise your manuscript. Please answer also the comments of both reviewers.

We would appreciate receiving your revised manuscript by Oct 19 2019 11:59PM. To enhance the reproducibility of your results, we recommend that if applicable you deposit your laboratory protocols in protocols.io, where a protocol can be assigned its own identifier (DOI) such that it can be cited independently in the future. For instructions see: http://journals.plos.org/plosone/s/submission-guidelines#loc-laboratory-protocols

We look forward to receiving your revised manuscript.

Kind regards,

Guillaume Souchay

Academic Editor

PLOS ONE

Journal Requirements:

Additional Editor Comments (if provided):

Reviewers' comments:

Reviewer's Responses to Questions

**Comments to the Author**

1. Is the manuscript technically sound, and do the data support the conclusions?

Reviewer #1: Yes

Reviewer #2: Yes

2. Has the statistical analysis been performed appropriately and rigorously? 

Reviewer #1: Yes

Reviewer #2: Yes

3. Have the authors made all data underlying the findings in their manuscript fully available?

Reviewer #1: Yes

Reviewer #2: Yes

4. Is the manuscript presented in an intelligible fashion and written in standard English?

Reviewer #1: Yes

Reviewer #2: Yes

5. Review Comments to the Author

Reviewer #1: This paper addresses a very interesting topic on impact of culling on the population dynamics of foxes. I enjoyed reading this paper. I don't have many criticisms or comments

My only concern is about the length of the manuscript. It is very detailed at the expense of clarity. I am a bit lost to find the most important take home messages.

I think the authors (and editor) should consider the option of making two consecutive papers, the first one concerning modelling (including simulation-estimation analysis and sensitivity analysis) and the second one focusing on the results and consequences on population dynamics of foxes in the 22 estates.

I think a specialist of state-space modelling, who I am not, should revised the paper. However, the modelling part is well described with precious appendices. Some of the appendices (S1, S2) include modelling of certain parameters used in the model, which are very important to consider and could be added to a first paper.

Methods concerning modelling are detailed in the methods section whereas results (simulation-estimation analysis S4 appendix and sensitivity analysis) are summarized in the Results section and put in S4 appendix. On the other hand, the Results section presented essentially parameter estimations, density reconstruction and population suppression on the 22 estates, while some of the parameters are not presented in the methods section.

I would recommend publication of the manuscript, after revision.

Yours sincerely

Specific comments:

Methods

Data

Fig. 1 Please shortly explain the difference between arable a) and arable b).

Could the names of the estates, which are used in the paper, be placed on this map?

L 139-140 Lamping should also not vary too much across the 22 estates so as to considering lamping effort per km² as an index of fox culling effort.

L160-162 The authors explained their choice of a two-weekly time-step but results from a weekly time-step are also presented L346 (for simulations, S4 appendix) and in the results section.

Table 1: Add ‘detectable’ to fox density Nt, as mentioned L 182-183

Population process model

L195-201 Please consider summarizing this paragraph so as to clearly define wt, without reading S1 appendix. Something like: ‘The Proportion of cubs on an estate weaned, wt, was calculated to ensure that cubs culled at earths within each year were not removed from the model before they had been produced. We used data from fox populations in SE England and Wales describing the probable conception dates of female foxes killed during pregnancy [2]. We fit a logistic distribution to describe the distribution of conception events in the female fox population over time and inferred a schedule of weaning.’

L206-207 The authors assumed that non culling mortality could be considered as a density-independent constant risk. This hypothesis is plausible if you consider road traffic collisions only but that’s not true for diseases.

L217 K was assumed to be constant… during the period in each estate?

L218-220 Consider putting these sentences in discussion.

Observation model

Why considering d and Et separately in Eq(6)? Does-it mean the rate of successful search in km².hr-1 is supposed to be constant for an estate over years?

Prior probability distributions for parameters

L260-265 Add symbols from table 1 for clarity in immigration rate.

Table 2: add ‘on a two-weekly step’ in the title

MCMC simulations

L297-301 Was reparametrisation only necessarily for MCMC simulations? I think the observation equation was also rewritten when fitting to an abundance index. If so, why not putting this part in the population process model (L221-242) section ?

Culling vs. non-culling mortality

L323 Define A, area

Results

Simulation-estimation analysis

Fig. B in this appendix is not readable.

L418-419 Fig. D in S4 appendix seems to show that Fig D this is not true for K and N0?

L421 Here, as a result, the estates that did not meet the data requirements should be given.

L422-441 It is not clear for me which estates should be excluded regarding requirements described.

Density reconstruction

L482 Replace maximum by minimum ?

Fig. 3 Fox density in the final time-step has never been defined?

L514-546 Appendices are indispensable here to understand.

Population suppression

How were pre-breeding and post-breeding fox density calculated?

Fig. 4 This figure should be revised. Maybe a histogram per estate with K, N(Feb) and N(Mar-Jul) besides would be more illustrative.

Mortality comparison

I don’t understand the interest of using cumulative mortalities (not defined in the Methods section)? How are these annual cumulative mortalities calculated?

L577-579 I don’t understand why a cumulative non-culling mortality exceeding carrying capacity after 4 years would imply a population turnover of four years in the absence of culling ?

Fig. 5 What is the scale for mortalities?

Sensitivity analyses

L 590 Why performing sensitivity analyses on six estates. This should be placed in the Methods section

I ‘m not sure Fig. 6 is very useful.

Fig. 7 Please precise a) and b)

Discussion

Parameter estimates

L677 The area YZM is also very small (4.3 km²). Is this very high immigration rate estimated on a two-week period?

Reviewer #2: The modelling approach is very clever and makes efficient use of the available data to uncover the underlying ecology of the system. The authors are to be congratulated on the innovative and thorough approach to analysis. The finding that fox populations on hunting estates with active culling programs are being propped up by immigration from surrounding non-culled areas is in many ways unsurprising, but it’s really pleasing to see this point being convincingly demonstrated using a very thorough analysis of the available data. I think this finding will be of interest to managers of predator populations generally, and not only to those managing foxes. Overall, the paper is very well written and was a pleasure to review.

A few specific comments and queries for consideration by the authors:

1. Line 240. Was any consideration given to using a mildly-informative prior for the process errors? I’ve had good results using priors with half-Cauchy and half-t with scale values set to a small value to reflect ecological plausibility when fitting state-space models and other kinds of hierarchical models to ecological data. Use of a sensitivity analysis is a good approach but may lead to over-precision in the estimation of some other parameters if uncertainty in the process errors isn’t admitted in the inferences.

2. Line 254. Overdispersion in number of fox detections could have a number of causes other than variation in lamping effort. For example, the efficiency with which foxes are detected could be influenced (for example) by weather, moonlight, habitat or observer skill/experience. No need to change the model to allow for these things (I suspect a more complex observation model will have identifiability problems), but perhaps briefly mention other potential drivers of overdispersion. The discussion of possible sources of variation in d in the Discussion around lines 730-739 touches on some of these influences, but doesn’t explicitly link them to overdispersion. Perhaps some brief mention in that part of the discussion would be worth considering.

3. Line 289. Perhaps “any value between 0 and K” would be more correct?

4. Line 308. Gelman-Rubin is great for assessing convergence, but I’d always recommend plotting some traces of key parameters to check for any oddities that G-R fails to identify.

5. Line 460. Just a query regarding model structure that occurred to me while examining Table 3. Is it possible that habitat or management variables (e.g. surrounding land use, abundance of prey species, availability of suitable sites for denning etc) might drive inter-site variation in carrying capacity? If this was the case including these as covariates on K might help with model identifiability. No need to update the model now, but is there any sign that estates with very high or low inferred values of K have systematic differences in known drivers of fox abundance? The possibility of estate-specific covariates informing local parameters values might be worthy of further exploration if the data will support it and could be of management relevance. I see some discussion of this issue with regard to the relationship between immigration rate and land-use in the Discussion, around line 681 which touches on this issue – perhaps a note suggesting the possibility of incorporating informative landscape covariates into the model could go here?

6. Caption for Figure 3, line 497. The timing of the bird nesting period in this part of the world won’t automatically be obvious to an international audience. Consider mentioning the dates in the caption or making dates-within-years easier to read of the x-axes of the time series plots – this would help readers interpret the seasonal timing of other data depicted on these plots as well.

7. The assessment of knock-down effects in Figure 4 is a good way of presenting these results. Was any consideration given to running the model using the estimated parameters, but assuming no culling effort (or alternatively higher culling effort). It should in principle be straightforward to generate posteriors of N under different management scenarios, which would be a more powerful means of assessing the effectiveness of the control operations.

8. The prior sensitivity analysis is very well done. It’s rare to see this done so thoroughly, and Figure 6 and 7 capture the results nicely.

9. Discussion, line 661. I have to gently take issue with the claim that the manuscript is the first application of state-space models to populations of red foxes. Scroggie et al. (2018) J. Appl. Ecol. 55: 2621-2631 used state-space models to study the dynamics of fox populations in Australia. (Full disclosure, I am the senior author of that paper).

10. Lines 75-759. Equating suppression of a predator population with success at protecting populations of prey is not always warranted. Relationships between predator abundance and prey mortality need not be linear, and suppression of the predator to a very low level might be needed to appreciably reduce prey mortality.

11. Lines 911-913. Genetic mark-recapture using scat samples might be one way around the problem of obtaining good non-cull mortality estimates which could be included in an IPM based on the present model. Genetic material from culled individuals could also be integrated into this framework.

6. PLOS authors have the option to publish the peer review history of their article (what does this mean?). If published, this will include your full peer review and any attached files.

Reviewer #1: No

Reviewer #2: Yes: Michael P. Scroggie

---

## [Author Response · Author response to Decision Letter 0]

11 Oct 2019

Please find detailed responses to each reviewer comment below (indented).

Review Comments to the Author

Reviewer #1:

This paper addresses a very interesting topic on impact of culling on the population dynamics of foxes. I enjoyed reading this paper. I don't have many criticisms or comments. My only concern is about the length of the manuscript. It is very detailed at the expense of clarity. I am a bit lost to find the most important take home messages. I think the authors (and editor) should consider the option of making two consecutive papers, the first one concerning modelling (including simulation-estimation analysis and sensitivity analysis) and the second one focusing on the results and consequences on population dynamics of foxes in the 22 estates. I think a specialist of state-space modelling, who I am not, should revised the paper. However, the modelling part is well described with precious appendices. Some of the appendices (S1, S2) include modelling of certain parameters used in the model, which are very important to consider and could be added to a first paper. Methods concerning modelling are detailed in the methods section whereas results (simulation-estimation analysis S4 appendix and sensitivity analysis) are summarized in the Results section and put in S4 appendix. On the other hand, the Results section presented essentially parameter estimations, density reconstruction and population suppression on the 22 estates, while some of the parameters are not presented in the methods section. I would recommend publication of the manuscript, after revision.

- When drafting the manuscript, we carefully considered splitting it into two as suggested by Reviewer #1. However, we ultimately decided that without the context and aim provided by fitting the model to real data, a simulation-estimation and sensitivity analysis paper would make for dull reading and not be sufficiently interesting to publish on its own. One option was to include data from one estate as a ‘case study’, but our view was that as it would be less likely to be published in that form, this would also seriously detract from the remaining manuscript and affect the publication of these results. As Reviewer #2 seemed happy with the structure, and as the editor did not recommend that we follow this suggestion, we have kept the original structure of the manuscript.

Specific comments:

Methods

Data

Fig. 1 Please shortly explain the difference between arable a) and arable b). Could the names of the estates, which are used in the paper, be placed on this map?

- Apologies, we had provided the incorrect Walsh & Harris (1996) reference. We have changed the reference to: Walsh, AL & Harris, S (1996) Foraging habitat preferences of vespertilionid bats in Britain. Journal of Applied Ecology 33: 508-518. Table 2 in that reference provides detail of the differences between the land class groupings. Adding further text into this manuscript would seem unnecessary.

- We regret that we are not able to put the estate codes onto this map, as given the estate area information contained in S1 Table the location of some of the estates could become identifiable, which would be a breach of confidentiality.

L 139-140 Lamping should also not vary too much across the 22 estates so as to considering lamping effort per km² as an index of fox culling effort.

- We are not entirely sure what is meant by this comment. If it relates to lamping behaviour by individual gamekeepers, we would agree that an inefficient gamekeeper conducting a given level of lamping effort might not have the same effect on the population than a more efficient gamekeeper. But this would be accounted for in the model by variation in the rate of successful search between gamekeepers. Any index of culling effort is imperfect, but given the prevalence of lamping, this one seems a defensible choice that provides information about variation in culling effort between estates.

L160-162 The authors explained their choice of a two-weekly time-step but results from a weekly time-step are also presented L346 (for simulations, S4 appendix) and in the results section.

- We argued that either a weekly or two-weekly time-step would be suitable. We used a weekly time-step for the bulk of the simulation-estimation analysis, but - given the greater computing time required when using a weekly time-step - we also examined the use of a two-weekly time-step. As reported in this section, the choice of timestep had a negligible effect on the results (posterior median estimates differed by <±5%), so to save computing time we used a two-weekly time-step when fitting the model to real data from the 22 estates.

Table 1: Add ‘detectable’ to fox density Nt, as mentioned L 182-183

- We have clarified Nt here as ‘Fox density excluding cubs prior to weaning’. A fox may be undetected for many reasons, being underage is just one of them. That is why we introduced it the phrase ‘detectable’ in quotes. On revising the manuscript, we felt ‘observable’ is a better phrase so have changed detectable to observable throughout.

Population process model

L195-201 Please consider summarizing this paragraph so as to clearly define wt, without reading S1 appendix. Something like: ‘The Proportion of cubs on an estate weaned, wt, was calculated to ensure that cubs culled at earths within each year were not removed from the model before they had been produced. We used data from fox populations in SE England and Wales describing the probable conception dates of female foxes killed during pregnancy [2]. We fit a logistic distribution to describe the distribution of conception events in the female fox population over time and inferred a schedule of weaning.’

- Some of this detail was in the previous paragraph to this one, but we have adopted this excellent suggestion and expanded this section to give a clearer summary description of how weaning schedule wt was derived.

L206-207 The authors assumed that non culling mortality could be considered as a density-independent constant risk. This hypothesis is plausible if you consider road traffic collisions only but that’s not true for diseases.

- We agree, but we feel confident making this assumption as epidemic diseases, e.g. mange, weren’t a feature of the rural fox population at that time. We now make this point in the revised text.

L217 K was assumed to be constant… during the period in each estate?

- Correct, we have now clarified.

L218-220 Consider putting these sentences in discussion.

- We have removed these sentences and put the most relevant detail earlier in this paragraph, to now say ‘In line with other authors [24,38,39], density-dependence in seasonal reproduction and/or immigration was modelled…’

Observation model

Why considering d and Et separately in Eq(6)? Does-it mean the rate of successful search in km².hr-1 is supposed to be constant for an estate over years?

- We have clarified that d is assumed to be constant for an estate over time. d is an intrinsic property of each gamekeeper and their working environment, whereas effort clearly varies from year to year. We did discuss the consequences of this assumption in the last section of the Discussion.

Prior probability distributions for parameters

L260-265 Add symbols from table 1 for clarity in immigration rate.

- We don’t understand this comment, as the symbols are already described in the ‘Population process model’ section and in Table 1. We have changed the order of parameter symbols to reflect the order in which the priors are described.

Table 2: add ‘on a two-weekly step’ in the title

- No, because that doesn’t apply to N0, K, r or d. Only v and M are defined on a two-weekly time-step.

MCMC simulations

L297-301 Was reparametrisation only necessarily for MCMC simulations? I think the observation equation was also rewritten when fitting to an abundance index. If so, why not putting this part in the population process model (L221-242) section ?

- As we describe, the reparameterization was performed only to improve the mixing speed of the Gibbs sampler and thereby cut down MCMC simulation time. Putting the reparameterised form of the state-space model, which includes both the state and observation equations, into the ‘Population process model’ section would lead to confusion as the reparameterised observation equation would come before it is initially presented in the ‘Observation model’ section. 

Culling vs. non-culling mortality

L323 Define A, area

- We thank the reviewer for raising this, as area should have been removed from Equation 13 due to Lt and St having been redefined in Table 1 in fox/km2 units. We have checked that this error was not in our code.

Results

Simulation-estimation analysis

Fig. B in this appendix is not readable.

- We are not sure what element is unreadable, in both the TIFF file uploaded and the PDF file created the figure seems perfectly readable to us. We chose to use red and blue for informative and vague prior reconstructions (and associated shading for the credible intervals) so there would not be issues with colour blindness. Perhaps something for the copyeditor?

L418-419 Fig. D in S4 appendix seems to show that Fig D this is not true for K and N0?

- This should have stated that true values were within the 95% credible intervals for all parameters, rather than the 80% CI. In re-examining the results, the statement was in fact true for K and the other parameters (although one population was close to the boundary of the 80% CI for K). In 80% of simulated populations (16/20) the true value of N0 was also within the 80% credible intervals, so we have revised this sentence to make this point.

L421 Here, as a result, the estates that did not meet the data requirements should be given.

- There seems no value in providing information relating to the excluded estates, as the data were not sufficient to use this model. In the ‘Data’ section we could have introduced only the sub-sample of estates we subsequently modelled, but we wanted to emphasise that there are minimum data requirements needed to reliably fit the model. Please also see response to next comment.

L422-441 It is not clear for me which estates should be excluded regarding requirements described.

- It appears that we weren’t clear enough that the excluded estates were not considered anywhere in this manuscript, as we only used data from 22 out of 74 estates that met the data requirements. We have amended the text to make this clearer. 

Density reconstruction

L482 Replace maximum by minimum ?

- The purpose of this sentence was to highlight the variability in detection rates between estates, using the maximum detection rate on each estate. To make it clearer, we have revised to ‘…with a maximum detection rate of 0.81 fox seen hr-1 on HIR compared to a maximum of >5 fox seen hr-1 on CHU, EWE, LEL, and NYP.’

Fig. 3 Fox density in the final time-step has never been defined?

- This means fox density in the final time-step of the data period. We did define this in the ‘MCMC simulations’ section of the Methods, as we used this final time-step to compare the post-model-pre-data distribution for fox density to the posterior distribution.

L514-546 Appendices are indispensable here to understand.

- The Appendices are of course needed to support the manuscript, but what is the alternative? Even if the manuscript were split into two (as per main comment), the simulation-estimation paper itself would still be needed as a reference.

Population suppression

How were pre-breeding and post-breeding fox density calculated?

- Pre-breeding fox density was calculated as the mean of the posterior median fox density during February, as detailed in the ‘Density reconstruction’ section. We have amended the text in this section to say ‘Fox density during the critical March-July gamebird nesting period (week 10-week 30) was calculated as the mean of the posterior median fox density during this time.’

Fig. 4 This figure should be revised. Maybe a histogram per estate with K, N(Feb) and N(Mar-Jul) besides would be more illustrative.

- Reviewer #2 seemed to like how the results were presented in Fig. 4, so we would prefer to keep it as is. We are not convinced that this suggestion would make the ‘knock-down’ effect any clearer. We have clarified in the caption for Fig. 4 that carrying capacity is specific to each estate.

Mortality comparison

I don’t understand the interest of using cumulative mortalities (not defined in the Methods section)? How are these annual cumulative mortalities calculated?

- The calculation of cumulative mortalities is described in the Methods, in the ‘Culling vs. non-culling mortality’ section. The cumulative mortality puts the magnitude of the cull into perspective relative to within-estate density, and the plotted line shows how quickly it is accumulated. The point at which mortality from culling exceeds within-estate density marks the point at which it is no longer plausible that the cull derives from within-estate production.

L577-579 I don’t understand why a cumulative non-culling mortality exceeding carrying capacity after 4 years would imply a population turnover of four years in the absence of culling ?

- Assuming that in the absence of culling the fox population of an estate was at carrying capacity, it seems intuitive to us that if the carrying capacity was some fixed number of foxes and that number of foxes had died from non-culling mortality factors after 4 years, there would be none of the original cohort left. Please correct us if our reasoning is wrong. No revisions have been made in response to this comment.

Fig. 5 What is the scale for mortalities?

- The scale is just a count of dead foxes, expressed as a density. We have clarified in the caption for Fig. 5, and in the relevant Methods section.

Sensitivity analyses

L 590 Why performing sensitivity analyses on six estates. This should be placed in the Methods section

- This was an arbitrary sample that represented estates covering a range of sizes and culling intensities. We couldn’t justify the time (or the space in the manuscript) required to conduct these analyses on all estates. We have moved this sentence to the Methods section and clarified our reasoning.

I ‘m not sure Fig. 6 is very useful.

- This is the evidence for any sensitivity of the model parameters to structural assumptions and prior distribution specifications. Reviewer #2 complimented us on how we had conducted our sensitivity analysis, so we do not want to remove this figure. We have now included more detail in the caption for Fig. 6, hopefully this makes its purpose clearer.

Fig. 7 Please precise a) and b)

- We have now included more detail about what is shown in each panel in the caption for Fig. 7. Panel (a) shows sensitivity to structural assumptions about the seasonality of immigration and the timing of cub recruitment. Panel (b) shows sensitivity to specification of prior distributions for M, r, K and N0. 

Discussion

Parameter estimates

L677 The area YZM is also very small (4.3 km²). Is this very high immigration rate estimated on a two-week period?

- We are not sure what the point of this comment is. We stated that this high estimate for v (which is defined on a two-week time-step in Table 1) was ‘equivalent’ to an annual number of foxes, i.e. multiplied by the 26 two-week time-steps in a year. In terms of the total number of foxes immigrating onto an estate, larger estates may have received more than on YZM (which was of average area). But that is why we compared densities, not numbers, as it is the rate of replacement that is of interest.

Reviewer #2:

The modelling approach is very clever and makes efficient use of the available data to uncover the underlying ecology of the system. The authors are to be congratulated on the innovative and thorough approach to analysis. The finding that fox populations on hunting estates with active culling programs are being propped up by immigration from surrounding non-culled areas is in many ways unsurprising, but it’s really pleasing to see this point being convincingly demonstrated using a very thorough analysis of the available data. I think this finding will be of interest to managers of predator populations generally, and not only to those managing foxes. Overall, the paper is very well written and was a pleasure to review.

- Thank you for these very kind comments, we are glad you enjoyed reading it.

A few specific comments and queries for consideration by the authors:

1. Line 240. Was any consideration given to using a mildly-informative prior for the process errors? I’ve had good results using priors with half-Cauchy and half-t with scale values set to a small value to reflect ecological plausibility when fitting state-space models and other kinds of hierarchical models to ecological data. Use of a sensitivity analysis is a good approach but may lead to over-precision in the estimation of some other parameters if uncertainty in the process errors isn’t admitted in the inferences.

- As you see, we did explore using informative priors for SD in the process errors, in the form of a log-normal prior, and presented these results as part of the sensitivity analysis in S2 Fig and S3 Fig. However, we had convergence problems when using this prior and do not think a weakly informative prior on process error SD would be more identifiable. We will explore the half-Cauchy and half-t options in future applications and have added a sentence into the Discussion referencing them.

2. Line 254. Overdispersion in number of fox detections could have a number of causes other than variation in lamping effort. For example, the efficiency with which foxes are detected could be influenced (for example) by weather, moonlight, habitat or observer skill/experience. No need to change the model to allow for these things (I suspect a more complex observation model will have identifiability problems), but perhaps briefly mention other potential drivers of overdispersion. The discussion of possible sources of variation in d in the Discussion around lines 730-739 touches on some of these influences, but doesn’t explicitly link them to overdispersion. Perhaps some brief mention in that part of the discussion would be worth considering.

- We agree, but think it makes more sense to mention these other factors here in the ‘Observation model’ section where we touch on the potential for overdispersion and where we say that we explored using the negative binomial distribution to model observation errors.

3. Line 289. Perhaps “any value between 0 and K” would be more correct?

- We agree and have made this change.

4. Line 308. Gelman-Rubin is great for assessing convergence, but I’d always recommend plotting some traces of key parameters to check for any oddities that G-R fails to identify.

- We had visually checked the MCMC trace plots; we just failed to report that we had done so. We have now added this detail into the sentence about convergence checking.

5. Line 460. Just a query regarding model structure that occurred to me while examining Table 3. Is it possible that habitat or management variables (e.g. surrounding land use, abundance of prey species, availability of suitable sites for denning etc) might drive inter-site variation in carrying capacity? If this was the case including these as covariates on K might help with model identifiability. No need to update the model now, but is there any sign that estates with very high or low inferred values of K have systematic differences in known drivers of fox abundance? The possibility of estate-specific covariates informing local parameters values might be worthy of further exploration if the data will support it and could be of management relevance. I see some discussion of this issue with regard to the relationship between immigration rate and land-use in the Discussion, around line 681 which touches on this issue – perhaps a note suggesting the possibility of incorporating informative landscape covariates into the model could go here?

- This is a very insightful comment, and indeed possible. We had already touched on the unavailability of a landscape covariate though in the Discussion about carrying capacity estimates. At the end of the Discussion section on culling effects on fox density, we had mentioned that gamebird release density might be responsible for some variation in fox density (as landscape alone is not a satisfactory predictor). The potential relationship with release density also extends to variation in parameter estimates, e.g. carrying capacity and immigration rate, and we have a manuscript in preparation on these relationships at different spatial scales. Unfortunately for the current manuscript, we did not have sufficient data from all estates to include release density as a covariate; if we had it could have been very informative. 

- To incorporate this comment, we have expanded on the Discussion around gamebird releasing to make the point that it ‘may create a ‘honeypot’ of higher prey availability, raising within-estate carrying capacity and possibly immigration rate’.

6. Caption for Figure 3, line 497. The timing of the bird nesting period in this part of the world won’t automatically be obvious to an international audience. Consider mentioning the dates in the caption or making dates-within-years easier to read of the x-axes of the time series plots – this would help readers interpret the seasonal timing of other data depicted on these plots as well.

- We agree this would make the plots clearer, so have changed the figure caption to say ‘…the bird nesting period (March-July) is shaded…’. We have also changed the captions of the similar figures for all other estates in S5 Appendix.

7. The assessment of knock-down effects in Figure 4 is a good way of presenting these results. Was any consideration given to running the model using the estimated parameters, but assuming no culling effort (or alternatively higher culling effort). It should in principle be straightforward to generate posteriors of N under different management scenarios, which would be a more powerful means of assessing the effectiveness of the control operations.

- We have considered this, as it is the focus of subsequent work where we have used the parameter estimates from this model within a management strategy evaluation framework. But in addition to further lengthening this manuscript, incorporating it here would add a predictive aspect, to what is currently purely descriptive.

8. The prior sensitivity analysis is very well done. It’s rare to see this done so thoroughly, and Figure 6 and 7 capture the results nicely.

- Many thanks, it is something we wish more authors would examine in sufficient detail.

9. Discussion, line 661. I have to gently take issue with the claim that the manuscript is the first application of state-space models to populations of red foxes. Scroggie et al. (2018) J. Appl. Ecol. 55: 2621-2631 used state-space models to study the dynamics of fox populations in Australia. (Full disclosure, I am the senior author of that paper).

- Many apologies for missing this, we have now added a reference to this paper but note that this application was not in the context of a culled (or exploited) population.

10. Lines 75-759. Equating suppression of a predator population with success at protecting populations of prey is not always warranted. Relationships between predator abundance and prey mortality need not be linear, and suppression of the predator to a very low level might be needed to appreciably reduce prey mortality.

- We are aware of this, but equally one can’t assume that partial suppression is insufficient to protect prey – estate DLQ illustrates this (lines 819-837 in the unmarked version of the revised manuscript). Our study is a carefully considered move to improve understanding one step at a time. Predator removal experiments (e.g. Tapper et al. 1996, Fletcher et al. 2010, referenced in our Introduction) have shown that in at least some circumstances suppression of common generalist predators including foxes can have positive effects on both breeding success and population trends of ground-nesting birds. As we have stated in the Introduction, such experiments are expensive, rarely done, and lack generality. It has been argued that predator control in general may not be as effective as those experiments, and particularly that where replacement of culled animals is rapid there will be no meaningful impact on predator density. Our study widens understanding of the effectiveness of fox control, from a couple of experiments in specific circumstances, to an appreciably broader sample of gamekeepers in a variety of circumstances, including those with low and high immigration. We have adjusted our text in various places to ensure this is clear.

11. Lines 911-913. Genetic mark-recapture using scat samples might be one way around the problem of obtaining good non-cull mortality estimates which could be included in an IPM based on the present model. Genetic material from culled individuals could also be integrated into this framework.

- We agree that would be fascinating to do, but the sampling and cost issues make it quite a challenging proposition. For completeness, we have incorporated the genetic identification option into this sentence. Whether it is a worthwhile investment given the relative scale of culling to non-culling mortality is debatable.

---

## [Editor Report · Decision Letter 1]

31 Oct 2019

Population dynamics of foxes during restricted-area culling in Britain: advancing understanding through state-space modelling of culling records

PONE-D-19-17263R1

Dear Dr. Porteus,

We are pleased to inform you that your manuscript has been judged scientifically suitable for publication and will be formally accepted for publication once it complies with all outstanding technical requirements.

With kind regards,

Guillaume Souchay

Academic Editor

PLOS ONE
---

## [Editor Report · Acceptance letter]

8 Nov 2019

PONE-D-19-17263R1 

Population dynamics of foxes during restricted-area culling in Britain: advancing understanding through state-space modelling of culling records 

Dear Dr. Porteus:

I am pleased to inform you that your manuscript has been deemed suitable for publication in PLOS ONE. Congratulations! Your manuscript is now with our production department. 

With kind regards,

on behalf of

Dr. Guillaume Souchay 

Academic Editor

PLOS ONE